# Exploring Algorithmic Fairness in Robust Graph Covering Problems

**Aida Rahmattalabi** [*]
rahmatta@usc.edu

**Phebe Vayanos** [*]
phebe.vayanos@usc.edu

**Anthony Fulginiti** [†]
anthony.fulginiti@du.edu

**Eric Rice** [*]
ericr@usc.edu

**Bryan Wilder** [‡]
bwilder@g.harvard.edu

**Amulya Yadav** [§]
amulya@psu.edu

**Milind Tambe** [‡]
milind_tambe@harvard.edu

## Abstract

Fueled by algorithmic advances, AI algorithms are increasingly being deployed in settings subject to unanticipated challenges with complex social effects. Motivated by real-world deployment of AI driven, social-network based suicide prevention and landslide risk management interventions, this paper focuses on robust graph covering problems subject to group fairness constraints. We show that, in the absence of fairness constraints, state-of-the-art algorithms for the robust graph covering problem result in biased node coverage: they tend to discriminate individuals (nodes) based on membership in traditionally marginalized groups. To mitigate this issue, we propose a novel formulation of the robust graph covering problem with group fairness constraints and a tractable approximation scheme applicable to real-world instances. We provide a formal analysis of the price of group fairness (PoF) for this problem, where we show that uncertainty can lead to greater PoF. We demonstrate the effectiveness of our approach on several real-world social networks. Our method yields competitive node coverage while significantly improving group fairness relative to state-of-the-art methods.

## 1 Introduction

**Motivation.** This paper considers the problem of selecting a subset of nodes (which we refer to as 'monitors') in a graph that can 'cover' their adjacent nodes. We are mainly motivated by settings where monitors are subject to failure and we seek to maximize worst-case node coverage. We refer to this problem as the *robust graph covering*. This problem finds applications in several critical real-world domains, especially in the context of optimizing social interventions on vulnerable populations. Consider for example the problem of designing *Gatekeeper training interventions for suicide prevention*, wherein a small number of individuals can be trained to identify warning signs of suicide among their peers [32]. A similar problem arises in the context of *disaster risk management in remote communities* wherein a moderate number of individuals are recruited in advance and trained to watch out for others in case of natural hazards (e.g., in the event of a landslide [40]). Previous research has shown that social intervention programs of this sort hold great promise [32, 40]. Unfortunately,

---

[*] University of Southern California

[†] University of Denver

[‡] Harvard University

[§] Pennsylvania State University

| Network Name | Network Size | Worst-case coverage of individuals by racial group (%) | | | | |
|---|---|---|---|---|---|---|
| | | White | Black | Hispanic | Mixed | Other |
| SPY1 | 95 | 70 | **36** | – | 86 | 94 |
| SPY2 | 117 | 78 | – | **42** | 76 | 67 |
| SPY3 | 118 | 88 | – | **33** | 95 | 69 |
| MFP1 | 165 | 96 | 77 | 69 | 73 | **28** |
| MFP2 | 182 | **44** | 85 | 70 | 77 | 72 |

Table 1: Racial discrimination in node coverage resulting from applying the algorithm in [45] on real-world social networks from two homeless drop-in centers in Los Angeles, CA [4], when 1/3 of nodes (individuals) can be selected as monitors, out of which at most 10% will fail. The numbers correspond to the worst-case percentage of covered nodes across all monitor availability scenarios.

in these real-world domains, intervention agencies often have very limited resources, e.g., moderate number of social workers to conduct the intervention, small amount of funding to cover the cost of training. This makes it essential to target the right set of monitors to cover a maximum number of nodes in the network. Further, in these interventions, the performance and availability of individuals (monitors) is *unknown* and *unpredictable*. At the same time, robustness is desired to guarantee high coverage even in worst-case settings to make the approach suitable for deployment in the open world.

Robust graph covering problems similar to the one we consider here have been studied in the literature, see e.g., [19, 45]. Yet, a major consideration distinguishes our problem from previous work: namely, the need for fairness. Indeed, when deploying interventions in the open world (especially in sensitive domains impacting life and death like the ones that motivate this work), care must be taken to ensure that algorithms do not discriminate among people with respect to protected characteristics such as race, ethnicity, disability, etc. In other words, we need to ensure that independently of their group, individuals have a high chance of being covered, a notion we refer to as *group fairness*.

To motivate our approach, consider deploying in the open world a state-of-the art algorithm for robust graph covering (which does not incorporate fairness considerations). Specifically, we apply the solutions provided by the algorithm from [45] on five real-world social networks. The results are summarized in Table 1 where, for each network, we report its size and the worst-case coverage by racial group. In all instances, there is significant disparity in coverage across racial groups. As an example, in network SPY1 36% of Black individuals are covered in the worst-case compared to 70% (resp. 86%) of White (resp. Mixed race) individuals. Thus, when maximizing coverage without fairness, (near-)optimal interventions end up mirroring any differences in degree of connectedness of different groups. In particular, well-connected groups at the center of the network are more likely to be covered (protected). Motivated by the desire to support those that are the less well off, we employ ideas from *maximin fairness* to improve coverage of those groups that are least likely to be protected.

**Proposed Approach and Contributions.** We investigate the *robust graph covering problem with fairness constraints*. Formally, given a social network, where each node belongs to a group, we consider the problem of selecting a subset of $I$ nodes (monitors), when at most $J$ of them may fail. When a node is chosen as a monitor and does not fail, all of its neighbors are said to be 'covered' and we use the term 'coverage' to refer to the total number of covered nodes. Our objective is to maximize worst-case coverage when any $J$ nodes may fail, while ensuring fairness in coverage across groups. We adopt maximin fairness from the Rawlsian theory of justice [41] as our fairness criterion: we aim to maximize the utility of the groups that are worse-off. To the best of our knowledge, ours is the first paper enforcing fairness constraints in the context of graph covering subject to node failure.

We make the following contributions: *(i)* We achieve maximin group fairness by incorporating constraints inside a robust optimization model, wherein we require that at least a fraction $W$ of each group is covered, in the worst-case; *(ii)* We propose a novel two-stage robust optimization formulation of the problem for which near-optimal conservative approximations can be obtained as a moderately-sized mixed-integer linear program (MILP). By leveraging the decomposable structure of the resulting MILP, we propose a Benders' decomposition algorithm augmented with symmetry breaking to solve practical problem sizes; *(iii)* We present the first study of price of group fairness (PoF), i.e., the loss in coverage due to fairness constraints in the graph covering problem subject to node failure. We provide upper bounds on the PoF for Stochastic Block Model networks, a widely

studied model of networks with community structure; *(iv)* Finally, we demonstrate the effectiveness of our approach on several real-world social networks of homeless youth. Our method yields competitive node coverage while significantly improving group fairness relative to state-of-the-art methods.

**Related Work.** Our paper relates to three streams of literature which we review.

*Algorithmic Fairness.* With increase in deployments of AI, OR, and ML algorithms for decision and policy-making in the open world has come increased interest in algorithmic fairness. A large portion of this literature is focused on resource allocation systems, see e.g., [13, 33, 50]. Group fairness in particular has been studied in the context of resource allocation problems [22, 42, 43]. A nascent stream of work proposes to impose fairness by means of constraints in an optimization problem, an approach we also follow. This is for example proposed in [1], and in [8, 24], and in [2] for machine learning, resource allocation, and matching problems, respectively. Several authors have studied the price of fairness. In [13], the authors provide bounds for maximin fair optimization problems. Their approach is restricted to convex and compact utility sets. In [6], the authors study price of fairness for indivisible goods with additive utility functions. In our graph covering problem, this property does not hold. Several authors have investigated notions of fairness under uncertainty, see e.g, [5, 28, 36, 50]. These papers all assume full distributional information about the uncertain parameters and cannot be employed in our setting where limited data is available about node availability. Motivated by data scarcity, we take a robust optimization approach to model uncertainty which does not require distributional information. This problem is highly intractable due to the combinatorial nature of both the decision and uncertainty spaces. When fair solutions are hard to compute, "approximately fair" solutions have been considered [33]. In our work, we adopt an approximation scheme. As such, our approach falls under the "approximately fair" category. Recently, several authors have emphasized the importance of fairness when conducting interventions in socially sensitive settings, see e.g., [3, 34, 44]. Our work most closely relates to [44], wherein the authors propose an algorithmic framework for fair influence maximization. We note that, in their work, nodes are not subject to failure and therefore their approach does not apply in our context.

*Submodular Optimization.* One can view the group-fair maximum coverage problem as a multi-objective optimization problem, with the coverage of each community being a separate objective. In the deterministic case, this problem reduces to the multi-objective submodular optimization problem [21], as coverage has the submodularity (diminishing returns) property. In addition, moderately sized problems of this kind can be solved optimally using integer programming technology. However, when considering uncertainty in node performance/availability, the objective function loses the submodularity property while exact techniques fail to scale to even moderate problem sizes. Thus, existing (exact or approximate) approaches do not apply. Our work more closely relates to the robust submodular optimization literature. In [19, 37], the authors study the problem of choosing a set of up to $I$ items, out of which $J$ fail (which encompasses as a special case the robust graph covering problem *without* fairness constraints). They propose a greedy algorithm with a constant (0.387) approximation factor, valid for $J = o(\sqrt{I})$, and $J = o(I)$, respectively. Finally, in [45], the authors propose another greedy algorithm with a general bound based on the curvature of the submodular function. These heuristics, although computationally efficient, are coverage-centered and do not take fairness into consideration. Thus, they may lead to discriminatory outcomes, see Table 1.

*Robust Optimization.* Our solution approach closely relates to the robust optimization paradigm which is a computationally attractive framework for obtaining equivalent or conservative approximations based on duality theory, see e.g., [7, 10, 49]. Indeed, we show that the robust graph covering problem can be written as a two-stage robust problem with binary second-stage decisions which is highly intractable in general [14]. One stream of work proposes to restrict the functional form of the recourse decisions to functions of benign complexity [12, 15]. Other works rely on partitioning the uncertainty set into finite sets and applying constant decision rules on each partition [15, 17, 31, 38, 47]. The last stream of work investigates the so-called $K$-adaptability counterpart [11, 20, 31, 39, 46], in which $K$ candidate policies are chosen in the first stage and the best of these policies is selected *after* the uncertain parameters are revealed. Our paper most closely relates to [31, 39]. In [31], the authors show that for bounded polyhedral uncertainty sets, linear two-stage robust optimization problems can be approximately reformulated as MILPs. Paper [39] extends this result to a special case of discrete uncertainty sets. We prove that we can leverage this approximation to reformulate robust graph covering problem with fairness constraints *exactly* for a much larger class of discrete uncertainty sets.

## 2 Fair and Robust Graph Covering Problem

We model a social network as a directed graph $\mathcal{G} = (\mathcal{N}, \mathcal{E})$, in which $\mathcal{N} := \{1, \ldots, N\}$ is the set of all nodes (individuals) and $\mathcal{E}$ is the set of all edges (social ties). A directed edge from $\nu$ to $n$ exists, i.e., $(\nu, n) \in \mathcal{E}$, if node $n$ can be covered by $\nu$. We use $\delta(n) := \{\nu \in \mathcal{N} : (\nu, n) \in \mathcal{E}\}$ to denote the set of neighbors (friends) of $n$ in $\mathcal{G}$, i.e., the set of nodes that can cover node $n$. Each node $n \in \mathcal{N}$ is characterized by a set of attributes (protected characteristics) such as age, race, gender, etc., for which fair treatment is important. Based on these node characteristics, we partition $\mathcal{N}$ into $C$ disjoint groups $\mathcal{N}_c$, $c \in \mathcal{C} := \{1, \ldots, C\}$, such that $\cup_{c \in \mathcal{C}} \mathcal{N}_c = \mathcal{N}$.

We consider the problem of selecting a set of $I$ nodes from $\mathcal{N}$ to act as 'peer-monitors' for their neighbors, given that the availability of each node is unknown a-priori and at most $J$ nodes may fail (be unavailable). We encode the choice of monitors using a binary vector $\boldsymbol{x}$ of dimension $N$ whose $n$th element is one iff the $n$th node is chosen. We require $\boldsymbol{x} \in \mathcal{X} := \{\boldsymbol{x} \in \{0,1\}^N : \mathbf{e}^\top \boldsymbol{x} \leq I\}$, where $\mathbf{e}$ is a vector of all ones of appropriate dimension. Accordingly, we encode the (uncertain) node availability using a binary vector $\boldsymbol{\xi}$ of dimension $N$ whose $n$th element equals one iff node $n$ does not fail (is available). Given that data available to inform the distribution of $\boldsymbol{\xi}$ is typically scarce, we avoid making distributional assumptions on $\boldsymbol{\xi}$. Instead, we view uncertainty as deterministic and set based, in the spirit of robust optimization [7]. Thus, we assume that $\boldsymbol{\xi}$ can take-on any value from the set $\Xi$ which is often referred to as the *uncertainty set* in robust optimization. The set $\Xi$ may for example conveniently capture failure rate information. Thus, we require $\boldsymbol{\xi} \in \Xi := \{\boldsymbol{\xi} \in \{0,1\}^N : \mathbf{e}^\top(\mathbf{e} - \boldsymbol{\xi}) \leq J\}$. A node $n$ is counted as 'covered' if at least one of its neighbors is a monitor and does not fail (is available). We let $\boldsymbol{y}_n(\boldsymbol{x}, \boldsymbol{\xi})$ denote if $n$ is covered for the monitor choice $\boldsymbol{x}$ and node availability $\boldsymbol{\xi}$.

$$\boldsymbol{y}_n(\boldsymbol{x}, \boldsymbol{\xi}) := \mathbb{I}\left(\sum_{\nu \in \delta(n)} \boldsymbol{\xi}_\nu \boldsymbol{x}_\nu \geq 1\right).$$

The coverage is then expressible as $F_{\mathcal{G}}(\boldsymbol{x}, \boldsymbol{\xi}) := \mathbf{e}^\top \boldsymbol{y}(\boldsymbol{x}, \boldsymbol{\xi})$. The *robust covering problem* which aims to maximize the worst-case (minimum) coverage under node failures can be written as

$$\max_{\boldsymbol{x} \in \mathcal{X}} \ \min_{\boldsymbol{\xi} \in \Xi} \ F_{\mathcal{G}}(\boldsymbol{x}, \boldsymbol{\xi}). \tag{$\mathcal{RC}$}$$

Problem ($\mathcal{RC}$) ignores fairness and may result in discriminatory coverage with respect to (protected) node attributes, see Table 1. We thus propose to augment the robust covering problem with fairness constraints. Specifically, we propose to achieve max-min fairness by imposing fairness constraints on each group's coverage: we require that at least a fraction $W$ of nodes from each group be covered. In [44], the authors show that by conducting a binary search for the largest $W$ for which fairness constraints are satisfied for all groups, the max-min fairness optimization problem is equivalent to the one with fairness constraints. Thus, we write the *robust covering problem with fairness constraints* as

$$\left\{ \max_{\boldsymbol{x} \in \mathcal{X}} \ \min_{\boldsymbol{\xi} \in \Xi} \ \sum_{c \in \mathcal{C}} F_{\mathcal{G},c}(\boldsymbol{x}, \boldsymbol{\xi}) \ : \ F_{\mathcal{G},c}(\boldsymbol{x}, \boldsymbol{\xi}) \geq W|\mathcal{N}_c| \quad \forall c \in \mathcal{C}, \ \forall \boldsymbol{\xi} \in \Xi \right\}, \tag{$\mathcal{RC}_{\text{fair}}$}$$

where $F_{\mathcal{G},c}(\boldsymbol{x}, \boldsymbol{\xi}) := \sum_{n \in \mathcal{N}_c} \boldsymbol{y}_n(\boldsymbol{x}, \boldsymbol{\xi})$ is the coverage of group $c \in \mathcal{C}$. Note that if $|\mathcal{C}| = 1$, Problem ($\mathcal{RC}_{\text{fair}}$) reduces to Problem ($\mathcal{RC}$), and if $\Xi = \{\mathbf{e}\}$, Problem ($\mathcal{RC}_{\text{fair}}$) reduces to the deterministic covering problem with fairness constraints. We emphasize that our approach can handle fairness with respect to more than one protected attribute by either: *(a)* partitioning the network based on joint values of the protected attributes and imposing a max-min fairness constraint for each group; or *(b)* imposing max-min fairness constraints for each protected attribute separately. Problem ($\mathcal{RC}_{\text{fair}}$) is computationally hard due to the combinatorial nature of both the uncertainty and decision spaces. Lemma 1 characterizes its complexity. Proofs of all results are in the supplementary document.

**Lemma 1.** *Problem ($\mathcal{RC}_{\text{fair}}$) is $\mathcal{NP}$-hard.*

## 3 Price of Group Fairness

In Section 2, we proposed a novel formulation of the robust covering problem incorporating fairness constraints, Problem ($\mathcal{RC}_{\text{fair}}$). Unfortunately, adding fairness constraints to Problem ($\mathcal{RC}$) comes at a price to overall worst-case coverage. In this section, we study this *price of group fairness*.

**Definition 1.** *Given a graph $\mathcal{G}$, the Price of Group Fairness $\mathrm{PoF}(\mathcal{G}, I, J)$ is the ratio of the coverage loss due to fairness constraints to the maximum coverage in the absence of fairness constraints, i.e.,*

$$\mathrm{PoF}(\mathcal{G}, I, J) := 1 - \frac{\mathrm{OPT}^{\mathrm{fair}}(\mathcal{G}, I, J)}{\mathrm{OPT}(\mathcal{G}, I, J)}, \tag{1}$$

*where $\mathrm{OPT}^{\mathrm{fair}}(\mathcal{G}, I, J)$ and $\mathrm{OPT}(\mathcal{G}, I, J)$ denote the optimal objective values of Problems ($\mathcal{RC}_{\mathrm{fair}}$) and ($\mathcal{RC}$), respectively, when $I$ monitors can be chosen and at most $J$ of them may fail.*

In this work, we are motivated by applications related to social networks, where it has been observed that people with similar (protected) characteristics tend to interact more frequently with one another, forming friendship groups (communities). This phenomenon, known as *homophily* [35], has been observed for characteristics such as race, gender, education, etc.[23]. This motivates us to study the PoF in Stochastic Block Model (SBM) networks [27], a widely accepted model for networks with community structure. In SBM networks, nodes are partitioned into $C$ disjoint communities $\mathcal{N}_c, c \in \mathcal{C}$. Within each community $c$, an edge between two nodes is present independently with probability $p_c^{\mathrm{in}}$. Between a pair of communities $c$ and $c' \in \mathcal{C}$, edges exist independently with probability $p_{cc'}^{\mathrm{out}}$ and we typically have $p_c^{\mathrm{in}} > p_{cc'}^{\mathrm{out}}$ to capture homophily. Thus, SBM networks are very adequate models for our purpose. We assume w.l.o.g. that the communities are labeled such that: $|\mathcal{N}_1| \leq \ldots \leq |\mathcal{N}_C|$.

**Deterministic Case.** We first study the PoF in the deterministic case for which $J = 0$. Lemma 2 shows that there are worst-case networks for which PoF can be arbitrarily bad.

**Lemma 2.** *Given $\epsilon > 0$, there exists a budget $I$ and a network $\mathcal{G}$ with $N \geq \frac{4}{\epsilon} + 3$ nodes such that $\mathrm{PoF}(\mathcal{G}, I, 0) \geq 1 - \epsilon$.*

Fortunately, as we will see, this pessimistic result is not representative of the networks that are seen in practice. We thus investigate the loss in expected coverage due to fairness constraints, given by

$$\overline{\mathrm{PoF}}(I, J) := 1 - \frac{\mathbb{E}_{\mathcal{G} \sim \mathrm{SBM}}[\mathrm{OPT}^{\mathrm{fair}}(\mathcal{G}, I, J)]}{\mathbb{E}_{\mathcal{G} \sim \mathrm{SBM}}[\mathrm{OPT}(\mathcal{G}, I, J)]}. \tag{2}$$

We emphasize that we investigate the loss in the expected coverage rather than the expected PoF for analytical tractability reasons. We make the following assumptions about SBM network.

**Assumption 1.** *For all communities $c \in \mathcal{C}$, the probability of an edge between two individuals in the community is inversely proportional to the size of the community, i.e., $p_c^{\mathrm{in}} = \Theta(|\mathcal{N}_c|^{-1})$.*

**Assumption 2.** *For any two communities $c, c' \in \mathcal{C}$, the probability of an edge between two nodes $n \in \mathcal{N}_c$ and $\nu \in \mathcal{N}_{c'}$ is $p_{cc'}^{\mathrm{out}} = O((|\mathcal{N}_c| \log^2 |\mathcal{N}_c|)^{-1})$.*

Assumption 1 is based on the observation that social networks are usually sparse. This means that most individuals do not form too many links, even if the size of the network is very large. Sparsity is characterized in the literature by the number of edges being proportional to the number of nodes which is the direct result of Assumption 1. Assumption 2 is necessary for meaningful community structure in the network. We now present results for the upper bound on $\overline{\mathrm{PoF}}$ in SBM networks.

**Proposition 1.** *Consider an SBM network model with parameters $p_c^{\mathrm{in}}$ and $p_{cc'}^{\mathrm{out}}$, $c, c' \in \mathcal{C}$, satisfying Assumptions 1 and 2. If $I = O(\log N)$, then*

$$\overline{\mathrm{PoF}}(I, 0) = 1 - \frac{\sum_{c \in \mathcal{C}} |\mathcal{N}_c|}{\sum_{c \in \mathcal{C}} |\mathcal{N}_c| d(C)/d(c)} - o(1), \text{ where } d(c) := \log |\mathcal{N}_c|(\log \log |\mathcal{N}_c|)^{-1}.$$

*Proof Sketch.* First, we show that under Assumption 1, the coverage within each community is the sum of the degrees of the monitoring nodes. Then, using the assumption on $I$ in the premise of the proposition (which can be interpreted as a "small budget assumption"), we evaluate the maximum coverage within each community. Next, we show that between-community coverage is negligible compared to within-community coverage. Thus, we determine the distribution of the monitors, in the presence and absence of fairness constraints. PoF is computed based on the these two quantities. ∎

**Uncertain Case.** Here, imposing fairness is more challenging as we do not know a-priori which nodes may fail. Thus, we must ensure that fairness constraints are satisfied under all failure scenarios.

**Proposition 2.** *Consider an SBM network model with parameters $p_c^{\text{in}}$ and $p_{cc'}^{\text{out}}$, $c, c' \in \mathcal{C}$, satisfying Assumptions 1 and 2. If $I = O(\log N)$, then*

$$\overline{\text{PoF}}(I, J) = 1 - \frac{\eta \sum_{c \in \mathcal{C}} |\mathcal{N}_c|}{(I - J) \times d(C)} - \frac{J \sum_{c \in \mathcal{C} \setminus \{C\}} d(c)}{(I - J) \times d(C)} - o(1),$$

*where $d(c)$ is as in Proposition 1 and $\eta := (I - CJ) \left( \sum_{c \in \mathcal{C}} |\mathcal{N}_c|/d(c) \right)^{-1}$.*

*Proof Sketch.* The steps of the proof are similar to those in the proof of Proposition 1 with the difference that, under uncertainty, monitors should be distributed such that the fairness constraints are satisfied even after $J$ nodes fail. Thus, we quantify a minimum number of monitors that should be allocated to each community. We then determine the worst-case coverage both in the presence and absence of fairness constraints. PoF is computed based on these two quantities. ∎

Propositions 1 and 2 show how PoF changes with the relative sizes of the communities for the deterministic and uncertain cases, respectively. Our analysis shows that without fairness, one should place all the monitors in the biggest community. Under a fair allocation however monitors are more evenly distributed (although larger communities still receive a bigger share). Figure 1 illustrates the PoF results in the case of two communities for different failure rates $\gamma$ ($J = \gamma I$), ignoring the $o(.)$ order terms. We keep the size of the first (smaller) community fixed and vary the size of the larger community. In both cases, if $|\mathcal{N}_1| = |\mathcal{N}_2|$, the PoF is zero since uniform distribution of monitors is optimal. As $|\mathcal{N}_2|$ increases, the PoF increases in both cases. Further increases in $|\mathcal{N}_2|$ result in a decrease in the PoF for the deterministic case: under a fair allocation, the bigger community receives a higher share of monitors which is aligned with the total coverage objective. Under uncertainty however, the PoF is non-decreasing: to guarantee fairness, additional monitors must be allocated to the smaller groups. This also explains why PoF increases with $\gamma$.

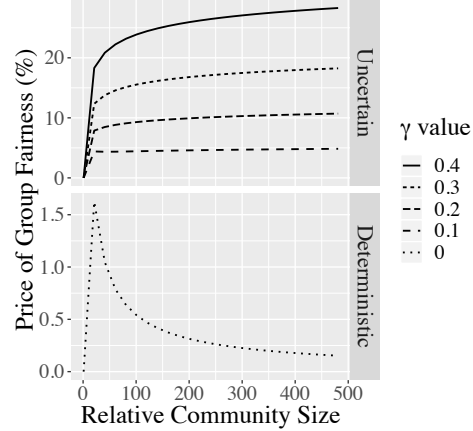

Figure 1: PoF in the uncertain (top) and deterministic (bottom) settings for SBM networks consisting of two communities ($\mathcal{C} = \{1, 2\}$) where the size of the first community is fixed at $|\mathcal{N}_1| = 20$ and the size of the other community is increased from $|\mathcal{N}_2| = 20$ to $10,000$. In the uncertain setting, $\gamma$ denotes the fraction of nodes that fail.

## 4 Solution Approach

Given the intractability of Problem ($\mathcal{RC}_{\text{fair}}$), see Lemma 1, we adopt a conservative approximation approach. To this end, we proceed in three steps. First, we note that a difficulty of Problem ($\mathcal{RC}_{\text{fair}}$) is the discontinuity of its objective function. Thus, we show that ($\mathcal{RC}_{\text{fair}}$) can be formulated equivalently as a *two-stage* robust optimization problem by introducing a *fictitious* counting phase *after* $\boldsymbol{\xi}$ is revealed. Second, we propose to approximate this decision made in the counting phase (which decides, for each node, whether it is or not covered). Finally, we demonstrate that the resulting approximate problem can be formulated equivalently as a moderately sized MILP, wherein the trade-off between suboptimality and tractability can be controlled by a single design parameter.

**Equivalent Reformulation.** For any given choice of $\boldsymbol{x} \in \mathcal{X}$ and $\boldsymbol{\xi} \in \Xi$, the objective $F_{\mathcal{G}}(\boldsymbol{x}, \boldsymbol{\xi})$ can be explicitly expressed as the optimal objective value of a covering problem. As a result, we can express ($\mathcal{RC}_{\text{fair}}$) equivalently as the two-stage *linear* robust problem

$$\max_{\boldsymbol{x} \in \mathcal{X}} \min_{\boldsymbol{\xi} \in \Xi} \max_{\boldsymbol{y} \in \mathcal{Y}} \left\{ \sum_{n \in \mathcal{N}} \boldsymbol{y}_n \ : \boldsymbol{y}_n \leq \sum_{\nu \in \delta(n)} \boldsymbol{\xi}_\nu \boldsymbol{x}_\nu, \ \forall n \in \mathcal{N} \right\}, \tag{3}$$

see Proposition 3 below. The second-stage binary decision variables $\boldsymbol{y} \in \mathcal{Y} := \{\boldsymbol{y} \in \{0,1\}^N : \sum_{n \in \mathcal{N}_c} \boldsymbol{y}_n \geq W|\mathcal{N}_c|, \ \forall c \in \mathcal{C}\}$ admit a very natural interpretation: at an optimal solution, $\boldsymbol{y}_n = 1$ if and only if node $n$ is covered. Henceforth, we refer to $\boldsymbol{y}$ as a *covering scheme*.

**Definition 2** (Upward Closed Set)**.** *A set $\mathcal{X}$ given as a subset of the partially ordered set $[0,1]^N$ equipped with the element-wise inequality, is said to be upward closed if for all $\boldsymbol{x} \in \mathcal{X}$ and $\bar{\boldsymbol{x}} \in [0,1]^N$ such that $\bar{\boldsymbol{x}} \geq \boldsymbol{x}$, it holds that $\bar{\boldsymbol{x}} \in \mathcal{X}$.*

Intuitively, sets involving lower bound constraints on the (sums of) parameters satisfy this definition. For example, sets that require a minimum fraction of nodes to be available. We can also consider group-based availability and require a minimum fraction of nodes to be available in every group.

**Assumption 3.** *We assume that: The set $\Xi$ is defined through $\Xi := \{0,1\}^N \cap \mathcal{T}$ for some upward closed set $\mathcal{T}$ given by $\mathcal{T} := \{\boldsymbol{\xi} \in \mathbb{R}^N : \boldsymbol{A}\boldsymbol{\xi} \geq \boldsymbol{b}\}$, with $\boldsymbol{A} \in \mathbb{R}^{R \times N}$ and $\boldsymbol{b} \in \mathbb{R}^R$.*

**Proposition 3.** *Problems ($\mathcal{RC}_{\text{fair}}$) and (3) are equivalent.*

$K$**-adaptability Counterpart.** Problem (3) has the advantage of being linear. Yet, its max-min-max structure precludes us from solving it directly. We investigate a conservative approximation to Problem (3) referred to as $K$-*adaptability counterpart*, wherein $K$ *candidate* covering schemes are chosen in the first stage and the best (feasible and most accurate) of those candidates is selected after $\boldsymbol{\xi}$ is revealed. Formally, the $K$-adaptability counterpart of Problem (3) is

$$\operatorname*{maximize}_{\substack{\boldsymbol{x} \in \mathcal{X} \\ \boldsymbol{y}^k \in \mathcal{Y}, \ k \in \mathcal{K}}} \quad \min_{\boldsymbol{\xi} \in \Xi} \max_{k \in \mathcal{K}} \left\{ \sum_{n \in \mathcal{N}} \boldsymbol{y}_n^k \ : \ \boldsymbol{y}_n^k \leq \sum_{\nu \in \delta(n)} \boldsymbol{\xi}_\nu \boldsymbol{x}_\nu \ \ \forall n \in \mathcal{N} \right\}, \tag{4}$$

where $\boldsymbol{y}^k$ denotes the $k$th candidate covering scheme, $k \in \mathcal{K}$. We emphasize that the covering schemes are not inputs but rather *decision variables* of the $K$-adaptability problem. Only the value $K$ is an input. The optimization problem will identify the best $K$ covering schemes that satisfy all the constraints including fairness constraints. The trade-off between optimality and computational complexity of Problem (4) can conveniently be tuned using the single parameter $K$.

**Reformulation as an MILP.** We derive an exact reformulation for the $K$-adaptability counterpart (4) of the *robust covering problem* as a moderately sized MILP. Our method extends the results from [39] to significantly more general uncertainty sets that are useful in practice, and to problems involving constraints on the set of covered nodes. Henceforth, we let $\mathcal{L} := \{0, \ldots, N\}^K$, and we define $\mathcal{L}_+ := \{\boldsymbol{\ell} \in \mathcal{L} : \boldsymbol{\ell} > \boldsymbol{0}\}$ and $\mathcal{L}_0 := \{\boldsymbol{\ell} \in \mathcal{L} : \boldsymbol{\ell} \not> \boldsymbol{0}\}$. We present a variant of the generic $K$-adaptability Problem (4), where the uncertainty set $\Xi$ is parameterized by vectors $\boldsymbol{\ell} \in \mathcal{L}$. Each $\boldsymbol{\ell}$ is a $K$-dimensional vector, whose $k$th component encodes if the $k$th covering scheme satisfies the constraints of the second stage maximization problem. In this case, $\boldsymbol{\ell}_k = 0$. Else, if the $k$th covering scheme is infeasible, $\boldsymbol{\ell}_k$ is equal to the index of a constraint that is violated.

**Theorem 1.** *Under Assumption 3, Problem (4) is equivalent to the mixed-integer bilinear program*

$$
\begin{aligned}
\max \quad & \tau \\
\text{s.t.} \quad & \tau \in \mathbb{R}, \ \boldsymbol{x} \in \mathcal{X}, \ \boldsymbol{y}^k \in \mathcal{Y} \ \forall k \in \mathcal{K} \\
& \left.\begin{array}{l}
\boldsymbol{\theta}(\boldsymbol{\ell}), \ \boldsymbol{\beta}^k(\boldsymbol{\ell}) \in \mathbb{R}_+^N, \ \boldsymbol{\alpha}(\boldsymbol{\ell}) \in \mathbb{R}_+^R, \ \boldsymbol{\nu}(\boldsymbol{\ell}) \in \mathbb{R}_+^K, \ \boldsymbol{\lambda}(\boldsymbol{\ell}) \in \Delta_K(\boldsymbol{\ell}) \\[4pt]
\tau \leq -\mathbf{e}^\top \boldsymbol{\theta}(\boldsymbol{\ell}) + \boldsymbol{\alpha}(\boldsymbol{\ell})^\top \boldsymbol{b} - \displaystyle\sum_{\substack{k \in \mathcal{K}: \\ \boldsymbol{\ell}_k \neq 0}} \left(\boldsymbol{y}_{\boldsymbol{\ell}_k}^k - 1\right) \boldsymbol{\nu}_k(\boldsymbol{\ell}) + \ldots \\[8pt]
\qquad \ldots + \displaystyle\sum_{\substack{k \in \mathcal{K}: \\ \boldsymbol{\ell}_k = 0}} \sum_{n \in \mathcal{N}} \boldsymbol{y}_n^k \boldsymbol{\beta}_n^k(\boldsymbol{\ell}) + \sum_{k \in \mathcal{K}} \boldsymbol{\lambda}_k(\boldsymbol{\ell}) \sum_{n \in \mathcal{N}} \boldsymbol{y}_n^k \\[8pt]
\boldsymbol{\theta}_n(\boldsymbol{\ell}) \leq \boldsymbol{A}^\top \boldsymbol{\alpha}(\boldsymbol{\ell}) + \displaystyle\sum_{\substack{k \in \mathcal{K}: \\ \boldsymbol{\ell}_k \neq 0}} \sum_{\nu \in \delta(\boldsymbol{\ell}_k)} \boldsymbol{x}_\nu \boldsymbol{\nu}_k(\boldsymbol{\ell}) - \sum_{\substack{k \in \mathcal{K}: \\ \boldsymbol{\ell}_k = 0}} \sum_{\nu \in \delta(n)} \boldsymbol{x}_\nu \boldsymbol{\beta}_n^k(\boldsymbol{\ell}) \ \ \forall n \in \mathcal{N}
\end{array}\right\} \forall \boldsymbol{\ell} \in \mathcal{L}_0 \\[10pt]
& \left.\begin{array}{l}
\boldsymbol{\theta}(\boldsymbol{\ell}) \in \mathbb{R}_+^N, \ \boldsymbol{\alpha}(\boldsymbol{\ell}) \in \mathbb{R}_+^R, \ \boldsymbol{\nu}(\boldsymbol{\ell}) \in \mathbb{R}_+^K \\[4pt]
1 \leq -\mathbf{e}^\top \boldsymbol{\theta}(\boldsymbol{\ell}) + \boldsymbol{\alpha}(\boldsymbol{\ell})^\top \boldsymbol{b} - \displaystyle\sum_{\substack{k \in \mathcal{K}: \\ \boldsymbol{\ell}_k \neq 0}} \left(\boldsymbol{y}_{\boldsymbol{\ell}_k}^k - 1\right) \boldsymbol{\nu}_k(\boldsymbol{\ell}) \\[8pt]
\boldsymbol{\theta}_n(\boldsymbol{\ell}) \leq \boldsymbol{A}^\top \boldsymbol{\alpha}(\boldsymbol{\ell}) + \displaystyle\sum_{\substack{k \in \mathcal{K}: \\ \boldsymbol{\ell}_k \neq 0}} \sum_{\nu \in \delta(\boldsymbol{\ell}_k)} \boldsymbol{x}_\nu \boldsymbol{\nu}_k(\boldsymbol{\ell}) \ \ \forall n \in \mathcal{N}
\end{array}\right\} \forall \boldsymbol{\ell} \in \mathcal{L}_+,
\end{aligned}
\tag{5}
$$

*which can be reformulated equivalently as an MILP using standard "Big-M" techniques since all bilinear terms are products continuous and binary variables. The size of this MILP scales with $|\mathcal{L}| = (N+1)^K$; it is polynomial in all problem inputs for any fixed $K$.*

*Proof Sketch.* The reformulation relies on three key steps: First, we partition the uncertainty set by using the parameter $\ell$. Next, we show that by relaxing the integrality constraint on the uncertain parameters $\boldsymbol{\xi}$, the problem remains unchanged. This is the key result that enables us to provide an equivalent formulation for Problem (4). Finally, we employ linear programming duality theory, to reformulate the robust optimization formulation over each subset. As a result, the formulation has two sets of decision variable: *(a)* The decision variables of the original problem; *(b)* Dual variables parameterized by $\ell$ which emerge from the dualization. ∎

**Bender's Decomposition.** In Problem (5), once binary variables $\boldsymbol{x}$ and $\{\boldsymbol{y}^k\}_{k \in \mathcal{K}}$ are fixed, the problem decomposes across $\ell$, i.e., all remaining variables are real valued and can be found by solving a linear program for each $\ell$. Bender's decomposition is an *exact* solution technique that leverages such decomposable structure for more efficient solution [9, 16]. Each iteration of the algorithm starts with the solution of a relaxed master problem, which is fed into the subproblems to identify violated constraints to add to the master problem. The process repeats until no more violated constraints can be identified. The formulations of master and subproblems are provided in Section E.

**Symmetry Breaking Constraints.** Problem (5) presents a large amount of symmetry. Indeed, given $K$ candidate covering schemes $\boldsymbol{y}^1, \dots, \boldsymbol{y}^K$, their indices can be permuted to yield another, distinct, feasible solution with identical cost. The symmetry results in significant slow down of the Brand-and-Bound procedure [18]. Thus, we introduce symmetry breaking constraints in the formulation (5) that stipulate the candidate covering schemes be lexicographically decreasing. We refer to [46] for details.

## 5 Computational Study on Social Networks of Homeless Youth

We evaluate our approach on the five social networks from Table 1. Details on the data are provided in Section A. We investigate the robust graph covering problem with maximin racial fairness constraints. All experiments were ran on a Linux 16GB RAM machine with Gurobi v6.5.0.

First, we compare the performance of our approach against the greedy algorithm of [45] and the degree centrality heuristic (DC). The results are summarized in Figure 2 (left). From the figure, we observe that an increase in $K$ results in an increase in performance along both axes, with a significant jump from $K = 1$ to $K = 2, 3$ (recall that $K$ controls complexity/optimality trade-off of our approximation). We note that the gain starts diminishing from $K = 2$ to $K = 3$. Thus, we only run up to $K = 3$. In addition the computational complexity of the problem increases exponentially with $K$, limiting us to increase $K$ beyond 3 for the considered instances. As demonstrated by our results, $K \sim 3$ was sufficient to considerably improve fairness of the covering at moderate price to efficiency. Compared to the baselines, with $K = 3$, we significantly improve the coverage of the worse-off group over greedy (resp. DC) by 11% (resp. 23%) on average across the five instances.

Second, we investigate the effect of uncertainty on the coverage of the worse-off group and on the PoF, for both the deterministic ($J = 0$) and uncertain ($J > 0$) cases as the number of monitors $I$ is varied in the set $\{N/3, N/5, N/7\}$. These settings are motivated by numbers seen in practice (typically, the number of people that can be invited is 15-20% of network size). Our results are summarized in Table 2. Indeed, from the table, we see for example that for $I = N/3$ and $J = 0$ our approach is able to improve the coverage of the worse-off group by 11-20% and for $J > 0$ the improvement in the worse-case coverage of the worse-off group is 7-16%. On the other hand, the PoF is very small: 0.3% on average for the deterministic case and at most 6.4% for the uncertain case. These results are consistent across the range of parameters studied. We note that the PoF numbers also match our analytical results on PoF in that uncertainty generally induces higher PoF.

Third, we perform a head-to-head comparison of our approach for $K = 3$ with the results in Table 1. Our findings are summarized in Table 5 in Section A. As an illustration, in SPY3, the worst-case coverage by racial group under our approach is: White 90%, Hispanic 44%, Mixed 85% and Other 87%. These numbers suggest that coverage of Hispanics (the worse-off group) has increased from

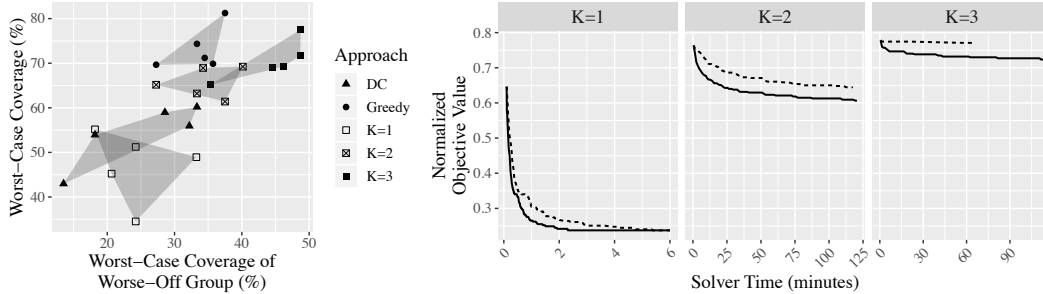

Figure 2: Left figure: Solution quality (overall worst-case coverage versus worst-case coverage of the group that is worse-off) for each approach (DC, Greedy, and $K$-adaptability for $K = 1, 2, 3$); The points represent the results of each approach applied to each of the five real-world social networks from Table 1; Each shaded area corresponds to the convex hull of the results associated with each approach; Approaches that are more fair (resp. efficient) are situated in the right- (resp. top-)most part of the graph. Right figure: Average of the ratio of the objective value of the master problem to the network size (across the five instances) in dependence of solver time for the Bender's decomposition approach (dotted line) and the Bender's decomposition approach augmented with symmetry breaking constraints (solid line). For both sets of experiments, the setting was $I = N/3$ and $J = 3$.

| Name | Size $N$ | Improvement in Min. Percentage Covered (%) | | | | | | PoF (%) | | | | | |
|---|---|---|---|---|---|---|---|---|---|---|---|---|---|
| | | Uncertainty Level $J$ | | | | | | Uncertainty Level $J$ | | | | | |
| | | 0 | 1 | 2 | 3 | 4 | 5 | 0 | 1 | 2 | 3 | 4 | 5 |
| SPY1 | 95 | 15 | 16 | 14 | 10 | 10 | 9 | 1.4 | 1.0 | 2.1 | 1.3 | 3.3 | 4.2 |
| SPY2 | 117 | 20 | 14 | 9 | 10 | 8 | 10 | 0.0 | 1.2 | 3.7 | 3.3 | 3.6 | 3.7 |
| SPY3 | 118 | 20 | 16 | 16 | 15 | 11 | 10 | 0.0 | 3.4 | 4.8 | 6.4 | 3.2 | 4.0 |
| MFP1 | 165 | 17 | 15 | 7 | 11 | 14 | 9 | 0.0 | 3.1 | 5.4 | 2.4 | 6.3 | 4.4 |
| MFP2 | 182 | 11 | 12 | 10 | 9 | 12 | 12 | 0.0 | 1.0 | 1.0 | 2.2 | 2.4 | 3.6 |
| Avg. ($I = N/3$) | | 16.6 | 14.6 | 11.2 | 11.0 | 11.0 | 10.0 | 0.3 | 1.9 | 3.4 | 3.1 | 3.8 | 4.0 |
| Avg. ($I = N/5$) | | 15.0 | 13.8 | 14.0 | 10.0 | 9.0 | 6.7 | 0.6 | 2.1 | 3.2 | 3.2 | 3.9 | 3.8 |
| Avg. ($I = N/7$) | | 12.2 | 11.4 | 11.2 | 11.4 | 8.2 | 6.4 | 0.1 | 2.5 | 3.5 | 3.2 | 3.5 | 4.0 |

Table 2: Improvement on the worst-case coverage of the worse-off group and associated PoF for each of the five real-world social networks from Table 1. The first five rows correspond to the setting $I = N/3$. In the interest of space, we only show averages for the settings $I = N/5$ and $I = N/7$. In the deterministic case ($J = 0$), the PoF is measured relative the coverage of the true optimal solution (obtained by solving the integer programming formulation of the graph covering problem). In the uncertain case ($J > 0$), the PoF is measured relative to the coverage of the greedy heuristic of [45].

33% to 44%, a significant improvement in fairness. To quantify the overall loss due to fairness, we also compute PoF values. The maximum PoF across all instances was at most 4.2%, see Table 5.

Finally, we investigate the benefits of augmenting our formulation with symmetry breaking constraints. Thus, we solve all five instances of our problem with the Bender's decomposition approach with and without symmetry breaking constraints. The results are summarized in Figure 2 (right). Across our experiments, we set a time limit of 2 hours since little improvement was seen beyond that. In all cases, and in particular for $K = 2$ and 3, symmetry breaking results in significant speed-ups. For $K = 3$ (and contrary to Bender's decomposition augmented with symmetry breaking), Bender's decomposition alone fails to solve the master problem to optimality within the time limit. We would like to remark that employing $K$-adaptability is necessary: indeed, Problem ($\mathcal{RC}_{\text{fair}}$) would not fit in memory. Similarly, using Bender's decomposition is needed: even for moderate values of $K$ (2 to 3), the $K$-adaptability MILP (5) could not be loaded in memory.

**Conclusion.** We believe that the robust graph covering problem with fairness constraints is worthwhile to investigate. It poses a huge number of challenges and holds great promise in terms of the realm of possible real-world applications with important potential societal benefits, e.g., to prevent suicidal ideation and death and to protect individuals during disasters such as landslides.

## Acknowledgements

We are grateful to three anonymous referees whose comments helped substantially improve the quality of this paper. This work was supported by the Smart & Connected Communities program of the National Science Foundation under NSF award No. 1831770 and by the US Army Research Office under grant number W911NF1710445.

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
