[Supplementary Material · Supplementary Material-FairCovering.pdf]

# A  Supplemental Material: Experimental Results in Section 5

**Data and Data Preprocessing.** The original datasets used throughout our paper are described in detail in [4]. They present 8 racial groups, with each individual belonging to a single group. To avoid misinterpretation of the results, we collect racial groups with a population $< 10\%$ of the network size $N$ under the "Other" category. The racial composition of the networks after the preprocessing is provided in Table 3. For instance, network SPY1 consists of $54\%$ White, $11\%$ Black, $15\%$ Mixed and $20\%$ Others. The empty entry for Hispanic indicates that their population was less than $10\%$; as a result, they are categorized under "Other".

| Network Name | White | Black | Hispanic | Mixed | Other |
|:---:|:---:|:---:|:---:|:---:|:---:|
| SPY1 | 54 | 11 | – | 15 | 20 |
| SPY2 | 55 | – | 11 | 21 | 13 |
| SPY3 | 58 | – | 10 | 18 | 14 |
| MFP1 | 16 | 38 | 22 | 16 | 8 |
| MFP2 | 16 | 32 | 22 | 20 | 10 |

Table 3: Racial composition (%) of the social networks considered after preprocessing

**Setting of Parameter $W$.** We now describe in detail the procedure we use to select $W$ in our experiments. As noted in Section 2, to achieve maximin fairness, $W$ must take the maximum value for which the problem is feasible (fairness constraints satisfied). Its value thus depends on other parameters, including $I$, $J$, and $K$. In our experiments, we conduct a search to identify the best value of $W$ for each setting. Specifically, we vary $W$ from 0 to 1, in increments of 0.04; we employ the largest $W$ for which the problem is feasible. By construction, this choice of $W$ guarantees that all of the fairness constraints are satisfied. In Table 4, we provide the values of $W$ associated with the results in Table 2 for $I = N/3$ and $K = 3$ and for each of the values of $J$.

| Network Name | $J=1$ | $J=2$ | $J=3$ | $J=4$ | $J=5$ |
|:---:|:---:|:---:|:---:|:---:|:---:|
| SPY1 | 0.44 | 0.40 | 0.36 | 0.32 | 0.32 |
| SPY2 | 0.56 | 0.52 | 0.48 | 0.44 | 0.36 |
| SPY3 | 0.44 | 0.36 | 0.32 | 0.28 | 0.24 |
| MFP1 | 0.52 | 0.48 | 0.44 | 0.40 | 0.32 |
| MFP2 | 0.56 | 0.52 | 0.44 | 0.40 | 0.32 |

Table 4: Values of $W$ output by our search procedure and used in the experiments associated with Table 2.

**Head-to-Head Comparison with Table 1.** We conduct a head-to-head comparison of our approach with the results from Table 1 which motivated our work. The results are summarized in Table 5. From the table we observe a consistent increase of 8-14% in worst-case coverage of the worse-off group. For example, in SPY3, the coverage of Hispanics has increased from 33% to 44%. We can also see that the PoF is moderate, ranging from 1-4.2%. The result for the MFP1 network suggests a 36% increase in the coverage of the "Other" group. We note that, by construction, this group consists of racial minorities with a population less than 10% of the network size. While this increase has impacted the coverage of "majority" groups, the worst-case coverage of the worse-off group has increased by 14% with a negligible PoF of 2.6%.

# B  Supplemental Material: Proof of Statements in Section 2

*Proof of Lemma 1.* For the special case when all monitors are available ($\Xi = \{\mathbf{e}\}$), there is a single community ($C = 1$), and no fairness constraints are imposed ($W = 0$), Problem ($\mathcal{RC}_{\text{fair}}$) reduces to the maximum coverage problem, which is known to be $\mathcal{NP}$-hard [26]. ∎

# C  Supplemental Material: Proofs of Statements in Section 3

In all of our analysis, we assume the graphs are undirected. This can be done without loss of generality and the results hold for directed graphs.

| Network Name | Network Size ($N$) | Worst-case coverage of individuals by racial group (%) | | | | | PoF (%) |
|---|---|---|---|---|---|---|---|
| | | White | Black | Hispanic | Mixed | Other | |
| SPY1 | 95 | 65 (70) | **45** (36) | – | 79 (86) | 88 (94) | 3.3 |
| SPY2 | 117 | 81 (78) | – | **50** (42) | 72 (76) | 73 (67) | 1.0 |
| SPY3 | 118 | 90 (88) | – | **44** (33) | 85 (95) | 87 (69) | 4.2 |
| MFP1 | 165 | 85 (96) | 69 (77) | 42 (69) | 73 (73) | **64** (28) | 2.6 |
| MFP2 | 182 | **56** (44) | 80 (85) | 70 (70) | 71 (77) | 72 (72) | 3.4 |

Table 5: Reduction in racial discrimination in node coverage resulting from applying our proposed algorithm relative to that of [45] on the five real-world social networks from Table 3, when 1/3 of nodes (individuals) can be selected as monitors, out of which at most 10% may fail. The numbers correspond to the worst-case percentage of covered nodes across all monitor availability scenarios. The numbers in the parentheses are solutions to the state-of-the-art algorithm [45] (same numbers as in Table 1).

(a) Original Graph      (b) With fairness      (c) Without fairness

Figure 3: Companion figure to Lemma 2. The figures illustrate a network sequence $\{\mathcal{G}_N\}_{N=5}^{\infty}$ parameterized by $N$ and consisting of two disconnected clusters: a small and a large one, with 4 and $N-4$ nodes, respectively. The small cluster remains intact as $N$ grows. The nodes in the large cluster form a clique. In the figures, each color (white, grey, black) represents a different group and we investigate the price of imposing fairness across these groups. The subfigures show the original graph (a) and an optimal solution when $I=2$ monitors can be selected in the cases (b) when fairness constraints are not imposed and (c) when fairness constraints are imposed, respectively. It holds that $\text{OPT}^{\text{fair}}(\mathcal{G}_N, 2, 0) = 4$ and $\text{OPT}(\mathcal{G}_N, 2, 0) = N - 3$ so that the PoF in $\mathcal{G}_N$ converges to one as $N$ tends to infinity.

## C.1 Worst-Case PoF

*Proof of Lemma 2.* Let $\{\mathcal{G}_N\}_{N=5}^{\infty}$ denote the graph sequence shown in Figure 3(a) (wherein all edges are bidirectional). The network consists of three groups (e.g., racial groups) for which fair treatment is important. Network $\mathcal{G}_N$ consists of two disjoint clusters: one involving four nodes and a bigger clique containing the remaining $(N-4)$ nodes. Suppose that we can choose $I=2$ nodes as monitors and that all of them are available ($J=0$). Observe that Problem ($\mathcal{RC}_{\text{fair}}$) is feasible only if $0 \leq W \leq (N-3)^{-1}$. For $W = (N-3)^{-1}$, the optimal solution places both nodes in the smaller cluster, see Figure 3(b). This way, at least one node from each group is covered. The total coverage for the fair solution is then equal to $\text{OPT}^{\text{fair}}(\mathcal{G}_N, 2, 0) = 4$. The maximum achievable coverage under no fairness constraints, however, is obtained by placing one monitor in each cluster, see Figure 3(c). Thus, the total coverage is equal to $\text{OPT}(\mathcal{G}_N, 2, 0) = N - 3$. As a result, $\text{PoF}(\mathcal{G}_N, 2, 0) = 1 - 4(N-3)^{-1}$ and for $N \geq 4/\epsilon + 3$, it holds that $\text{PoF}(\mathcal{G}_N, 2, 0) \geq 1 - \epsilon$. The proof is complete. ∎

## C.2 Supporting Results for the PoF Derivation

In this section, we provide the preliminary results needed in the derivation of the PoF for both the deterministic and robust graph covering problems. First, we provide two results (Lemmas 2 and 3) from the literature which characterize the maximum degree, as well as the expected number of maximum-degree nodes in sparse Erdős Rény graphs [25, 30]. We note that in SBM graphs which are used in our PoF analysis, each community $c \in \mathcal{C}$, when viewed in isolation, is an instance of the Erdős Rényi graph, in which each edge exists independently with probability $p_c^{\text{in}}$. These results are useful to evaluate the coverage of each community $c \in \mathcal{C}$ under the sparsity Assumption 1. Specifically,

they enable us to show in Lemma 4 that, in sparse Erdős Rényi graphs, the coverage can be evaluated approximately as the sum of the degrees of the monitoring nodes. Thus, the maximum coverage within each community in an SBM network can obtained by selecting the maximum degree nodes. Lastly, we prove Lemma 6 which will be useful to show that coverage from monitoring nodes in other communities in SBM networks is negligible.

In what follows, we use $\mathbb{G}_{N,p}$ to denote a random instance of Erdős Rény graphs on vertex set $\mathcal{N}(= \{1, \ldots, N\})$, where each edge occurs independently with probability $p$. Following the notational conventions in [29], we will say that a sequence of events $\{\mathbb{A}_n\}_{n=1}^N$ occurs with high probability if $\lim_{n \to \infty} \mathbb{P}(\mathbb{A}_n) = 1$ and, given a graph $\mathcal{G}$, we let $\Delta(\mathcal{G})$, the maximum degree of vertices of $\mathcal{G}$.

**Theorem 2** ([29, Theorem 3.4]). *Let $\{\mathbb{G}_{N,p}\}_{N=1}^{\infty}$ a sequence of graphs. If $p = \Theta(N^{-1})$, then with high probability*

$$\lim_{N \to \infty} \Delta(\mathbb{G}_{N,p}) = \frac{\log N}{\log \log N}.$$

**Lemma 3.** *Let $\{\mathbb{G}_{N,p}\}_{N=1}^{\infty}$ a sequence of graphs with $p = \Theta(N^{-1})$. Let $\sigma(N) := \log N (\log \log N)^{-1}$. Then, it holds that*

$$\mathbb{E}[X_{\sigma(N)}(\mathbb{G}_{N,p})] \geq N^{\frac{\log \log \log N - o(1)}{\log \log N}},$$

*where $X_{\sigma(N)}(\mathbb{G}_{N,p})$ is the number of vertices of degree $\sigma(N)$ in $\mathbb{G}_{N,p}$.*

**Proof.** We borrow results from [29, Theorem 3.4], where the authors show that

$$\mathbb{E}[X_{\sigma(N)}(\mathbb{G}_{N,p})] = \exp\left(\frac{\log N}{\log \log N}(\log \log \log N - o(1)) + O\left(\frac{\log N}{\log \log N + 2 \log \log \log N}\right)\right),$$

We further simplify the expression in Lemma 3 by eliminating the $O(.)$ term and we obtain

$$\mathbb{E}[X_{\sigma(N)}(\mathbb{G}_{N,p})] \geq N^{\frac{\log \log \log N - o(1)}{\log \log N}},$$

$\blacksquare$

Lemma 3 ensures that our budget for selecting monitors $I = O(\log N)$, is (asymptotically) smaller than number of nodes with degree $\Delta(\mathbb{G}_{N,p})$.

**Lemma 4.** *Let $\{\mathbb{G}_{N,p}\}_{N=1}^{\infty}$ be a sequence of graphs with $p = \Theta(N^{-1})$. Suppose that the number of monitors is $I = O(\log N)$. Then, for all $\nu$, there exists a graph $\mathbb{G}_{N,p}$ such that the difference between the expected maximum coverage in $\mathbb{G}_{N,p}$ and the expected number of neighbors of the monitoring nodes is bounded. Precisely, if $\boldsymbol{x}(\mathbb{G}_{N,p})$ is the indicator vector of the highest degree nodes in $\mathbb{G}_{N,p}$, we have*

$$\sum_{n \in \mathcal{N}} \mathbb{E}\left[\boldsymbol{x}_n(\mathbb{G}_{N,p}) | \delta_{\mathbb{G}_{N,p}}(n)|\right] - \mathbb{E}\left[F_{\mathbb{G}_{N,p}}(\boldsymbol{x}(\mathbb{G}_{N,p}), \mathbf{e})\right] \leq \nu,$$

*where $\delta_{\mathbb{G}_{N,p}}(n)$ is the set of neighbors of $n$ in $\mathbb{G}_{N,p}$ and $\nu$ is the error term and it is $\nu = o(1)$.*

**Proof.** Let $Y_n$ be the event that node $n$ is covered. Also, let $Z_n^i$ the event that node $n$ is covered by the $i$th highest degree node (and by potentially other nodes too). Without loss of generality, assume that the nodes with lower indexes have higher degrees, i.e., $|\delta(1)| \geq \cdots \geq |\delta(N)|$. The probability that node $n$ is covered can be written as

$$\mathbb{P}(Y_n) = \mathbb{P}\left(\cup_{i=1}^I Z_n^i\right). \tag{6}$$

From the Bonferroni inequalities, we have

$$\mathbb{P}(\cup_{i=1}^I Z_n^i) \geq \sum_{i=1}^I \left(\mathbb{P}(Z_n^i) - \sum_{j=i}^I \mathbb{P}(Z_n^i \cap Z_n^j)\right) \tag{7}$$

and

$$\mathbb{P}(\cup_{i=1}^I Z_n^i) \ \leq \ \sum_{i=1}^I \mathbb{P}(Z_n^i). \tag{8}$$

Define $Y := \sum_{i=1}^N Y_n$ as the (random) total coverage. With a slight abuse of notation, we view $Y_n$ and $Z_n^i$ as Bernoulli random binary variables that are equal to 1 if and only if the associated event occurs. As a result, we can substitute the probability terms with their expected values. Combining Equations (6), (7) and (8), we obtain

$$\sum_{i=1}^I \left( \mathbb{E}[Z_n^i] - \sum_{j=i}^I \mathbb{E}[Z_n^i Z_n^j] \right) \ \leq \ \mathbb{E}[Y_n] \ \leq \ \sum_{i=1}^I \mathbb{E}[Z_n^i], \quad \forall n \in \mathcal{N},$$

where we used the fact that $\mathbb{P}(Z_n^i \cap Z_n^j) = \mathbb{P}(Z_n^i)\mathbb{P}(Z_n^j) = \mathbb{E}(Z_n^i)\mathbb{E}(Z_n^j) = \mathbb{E}(Z_n^i Z_n^j)$ by independence of the events $Z_n^i$ and $Z_n^j$. Summing over all $n$ yields

$$\sum_{n \in \mathcal{N}} \left( \sum_{i=1}^I \mathbb{E}[Z_n^i] - \sum_{j=i}^I \mathbb{E}[Z_n^i Z_n^j] \right) \ \leq \ \sum_{n \in \mathcal{N}} \mathbb{E}[Y_n] \ \leq \ \sum_{n \in \mathcal{N}} \sum_{i=1}^I \mathbb{E}[Z_n^i].$$

Changing the order of the summations, it follows that

$$\sum_{i=1}^I \left( \sum_{n \in \mathcal{N}} \mathbb{E}[Z_n^i] - \sum_{j=i}^I \sum_{n \in \mathcal{N}} \mathbb{E}[Z_n^i Z_n^j] \right) \ \leq \ \mathbb{E}[Y] \ \leq \ \sum_{i=1}^I \sum_{n \in \mathcal{N}} \mathbb{E}[Z_n^i],$$

where we have used $\mathbb{E}[Y] = \sum_{i=1}^N \mathbb{E}[Y_n]$. By definition of $\delta_{\mathbb{G}_{N,p}}(i)$, since $x_i(\mathbb{G}_{N,p}) = 1$ for $i = 1, \ldots, I$, it holds that the number of nodes covered by node $i$, $\sum_{n \in \mathcal{N}} \mathbb{E}[Z_n^i] = \mathbb{E}[|\delta_{\mathbb{G}_{N,p}}(i)|]$. Also, we remark that $\mathbb{E}[Y] = \mathbb{E}[F_{\mathbb{G}_{N,p}}(\boldsymbol{x}(\mathbb{G}_{N,p}), \mathbf{e})]$. Thus, the above sequence of inequalities is equivalent to

$$\sum_{i=1}^I \left( \mathbb{E}[|\delta_{\mathbb{G}_{N,p}}(i)|] - \sum_{j=i}^I \sum_{n \in \mathcal{N}} \mathbb{E}[Z_n^i Z_n^j] \right) \ \leq \ \mathbb{E}[F_{\mathbb{G}_{N,p}}(\boldsymbol{x}(\mathbb{G}_{N,p}), \mathbf{e})] \ \leq \ \sum_{i=1}^I \mathbb{E}[|\delta_{\mathbb{G}_{N,p}}(i)|],$$

where, by reordering terms, we obtain

$$0 \ \leq \ \sum_{i=1}^I \mathbb{E}[|\delta_{\mathbb{G}_{N,p}}(i)|] - \mathbb{E}[F_{\mathbb{G}_{N,p}}(\boldsymbol{x}(\mathbb{G}_{N,p}), \mathbf{e})] \ \leq \ \sum_{i=1}^I \sum_{j=i}^I \sum_{n \in \mathcal{N}} \mathbb{E}[Z_n^i Z_n^j].$$

Note that $\mathbb{E}[\boldsymbol{x}_n(\mathbb{G}_{N,p})] = 1, \forall n \leq I$ since by assumption the nodes are ordered by decreasing order of their degree, so the nodes indexed from 1 to $I$ are selected in each realization of the graph. Thus,

$$\begin{aligned} \sum_{i=1}^I \mathbb{E}[|\delta_{\mathbb{G}_{N,p}}(i)|] \ &= \ \sum_{n \in \mathcal{N}} \mathbb{E}\left[\boldsymbol{x}_n(\mathbb{G}_{N,p})\right] \mathbb{E}\left[|\delta_{\mathbb{G}_{N,p}}(n)|\right] \\ &= \ \sum_{n \in \mathcal{N}} \mathbb{E}\left[\boldsymbol{x}_n(\mathbb{G}_{N,p})|\delta_{\mathbb{G}_{N,p}}(n)|\right], \end{aligned}$$

which yields

$$\sum_{n \in \mathcal{N}} \mathbb{E}\left[\boldsymbol{x}_n(\mathbb{G}_{N,p})|\delta_{\mathbb{G}_{N,p}}(n)|\right] - \mathbb{E}[F_{\mathbb{G}_{N,p}}(\boldsymbol{x}(\mathbb{G}_{N,p}), \mathbf{e})] \ \leq \ \sum_{i=1}^I \sum_{j=i}^I \sum_{n \in \mathcal{N}} \mathbb{E}[Z_n^i Z_n^j]. \tag{9}$$

The right-hand side of Equation (9) is the error term and we denote it by $\nu = \sum_{i=1}^I \sum_{j=i}^I \sum_{n \in \mathcal{N}} \mathbb{E}[Z_n^i Z_n^j]$. This error term determines the difference between the true value of the coverage and the expected sum of the degrees of the monitoring nodes. Given that $p = \Theta(N^{-1})$, we can precisely evaluate the error term. First, we note that since in the Erdős-Rényi model edges are

drawn independently, we can write $\mathbb{E}[Z_n^i Z_n^j] = \mathbb{E}[Z_n^i]\,\mathbb{E}[Z_n^j]$. Using Theorem 2 and Lemma 3, and given that the monitors are the highest degree nodes in any realization of the graph, we can write

$$\mathbb{E}[Z_n^i] = \mathbb{E}[Z_n^j] = \Theta\left(\frac{1}{N}\frac{\log N}{\log\log N}\right).$$

We thus obtain

$$\nu = \Theta\left(\frac{I^2}{N}\left(\frac{\log N}{\log\log N}\right)^2\right).$$

By the assumption on the order of $I$, it follows that $\lim_{N\to\infty}\nu = 0$, which concludes the proof. ∎

We now prove the following lemma which will be used in proof of the subsequent results.

**Lemma 5.** *Let $X_i$ for $i = 1,\ldots,Q$ be $Q$ i.i.d samples from normal distribution with mean $\mu$ and standard deviation $\sigma$. Also, let $Z = \max_{i\in\{1,\cdots,Q\}} X_i$. It holds that*

$$\mathbb{E}[Z] \le \mu + \sigma\sqrt{2\log Q}.$$

**Proof.** By Jensen's inequality,

$$
\begin{aligned}
\exp(t\,\mathbb{E}[Z]) \le \mathbb{E}[\exp(tZ)] &= \mathbb{E}[\exp(t\max_{i=1,\ldots,Q} X_i)]\\
&\le \sum_{i=1}^{Q}\mathbb{E}[\exp(tX_i)]\\
&= Q\exp(\mu t + t^2\sigma^2/2),
\end{aligned}
$$

where the last equality follows from the definition of the Gaussian moment generating function. Taking the logarithm of both sides of this inequality, we can obtain

$$\mathbb{E}[Z] \le \mu + \frac{\log Q}{t} + \frac{t\sigma^2}{2}.$$

For the tightest upper-bound, we set $t = \sqrt{2\log Q}/\sigma$. Thus, we obtain

$$\mathbb{E}[Z] \le \mu + \sigma\sqrt{2\log Q}.$$

∎

**Lemma 6.** *Consider $\mathbb{B}_{N,M,p}$ to be a random instance of a bipartite graph on the vertex set $\mathcal{N} = \mathcal{L}\cup\mathcal{R}$, where $N = |\mathcal{R}\cup\mathcal{L}|$ and $M := |\mathcal{R}|$ and $p = O\left((M\log^2 M)^{-1}\right)$ is the probability that each edge exists (independently). Suppose that monitoring nodes can only be chosen from the set $\mathcal{L}$ and that at most $I$ monitors can be selected. Then, it holds that*

$$\mathbb{E}\left[\max_{\substack{\boldsymbol{x}\in\{0,1\}^{|\mathcal{L}|}:\\ \sum_{n\in\mathcal{L}} x_n = I}} F_{\mathbb{B}_{N,M,p}}(\boldsymbol{x},\mathbf{e})\right] = IO\left(\frac{1}{\log^2 M}\right).$$

**Proof.** We note that the degree of node $i$, $\delta_{\mathbb{B}_{N,M,p}}(i)$, follows a binomial distribution with mean $Mp$. Given we are interested in $N, M\to\infty$, we can approximate the binomial distribution with a normal distribution [48] with mean $Mp$ and standard deviation $\sqrt{Mp(1-p)}$. Using the result of Lemma 5, we obtain

$$\mathbb{E}[\Delta_{\mathbb{B}_{N,M,p}}] = O\left(Mp + \sqrt{Mp(1-p)}\sqrt{2\log(N-M)}\right) = O(Mp).$$

Using the above result combined with the assumption on $p$, we can bound the expected maximum degree of $\mathcal{B}$.

$$\mathbb{E}[\Delta_{\mathbb{B}_{N,M,p}}] = O\left(\frac{1}{\log^2 M}\right).$$

As a result, the maximum expected coverage of the $I$ monitoring nodes is upper-bounded as

$$\mathbb{E}\left[\max_{\substack{\boldsymbol{x}\in\{0,1\}^{N}:\\ \sum_{n\in\mathcal{L}} x_n = I}} F_{\mathbb{B}_{N,M,p}}(\boldsymbol{x},\mathbf{e})\right] \le I\,\mathbb{E}[\Delta_{\mathbb{B}_{N,M,p}}] = IO\left(\frac{1}{\log^2 M}\right).$$

and the proof is complete. ∎

## C.3 PoF in the Deterministic Case

Next, we prove the main result which is the derivation of the PoF for the deterministic graph covering problem. The idea of the proof is as follows: by Lemmas 3 and 4, we are able to evaluate the coverage of each community. By Lemma 6, we upper bound the between-community coverage. In other words, based on Lemma 6, we conclude that in every instance of the coverage problem, the between-community coverage is zero (asymptotically) with high probability. Thus, the allocation of monitoring nodes is only dependant on the within-community coverage. Using this observation, we can determine the allocation of the monitors both in the presence and absence of fairness constraints. Subsequently, we are able to evaluate the coverage in both cases. PoF can be then computed based on these two quantities, see Equation (2).

*Proof of Proposition 1.* Let $\mathbb{S}_N$ be a random instance of the SBM network with size $N$. Consider $\boldsymbol{s}(\mathbb{S}_N) \in \mathbb{Z}^C$ to be the number of allocated monitoring nodes to each of the $C$ communities, i.e., $\boldsymbol{s}_c(\mathbb{S}_N) = \sum_{n \in \mathcal{N}_c} \boldsymbol{x}_n(\mathbb{S}_N)$. Using the result of Lemmas 4 and 6, we can measure the expected maximum coverage as

$$\lim_{N \to \infty} \mathbb{E}[\text{OPT}(\mathbb{S}_N, I, 0)] = \lim_{N \to \infty} \mathbb{E}\left[\max_{\boldsymbol{x}(\mathbb{S}_N) \in \mathcal{X}} F_{\mathbb{S}_N}(\boldsymbol{x}, \mathbf{e})\right] = \mathbb{E}\left[\lim_{N \to \infty} \max_{\boldsymbol{x}(\mathbb{S}_N) \in \mathcal{X}} F_{\mathbb{S}_N}(\boldsymbol{x}, \mathbf{e})\right],$$

where the last equality is obtained by exchanging the expectation and limit. Using Lemma 2 and since the maximum degree is convergent to $d(c)$, we can exchange the limit and maximization term. Thus, we will have

$$
\begin{aligned}
\mathbb{E}\left[\lim_{N \to \infty} \max_{\boldsymbol{x}(\mathbb{S}_N) \in \mathcal{X}} F_{\mathbb{S}_N}(\boldsymbol{x}, \mathbf{e})\right] &= \mathbb{E}\left[\max_{\boldsymbol{x}(\mathbb{S}_N) \in \mathcal{X}} \lim_{N \to \infty} F_{\mathbb{S}_N}(\boldsymbol{x}, \mathbf{e})\right] \\
&= \mathbb{E}\left[\max_{\boldsymbol{s}(\mathbb{S}_N) \in \mathbb{Z}^C} \sum_{c \in \mathcal{C}} \boldsymbol{s}_c(\mathbb{S}_N) d(c) + o(1)\right],
\end{aligned}
$$

which given that $d(c)$ is only dependent on the size of the communities in $\mathbb{S}_N$ is equivalent to

$$\lim_{N \to \infty} \mathbb{E}[\text{OPT}(\mathbb{S}_N, I, 0)] = \max_{\boldsymbol{s}(\mathbb{S}_N)} \sum_{c \in \mathcal{C}} \boldsymbol{s}_c(\mathbb{S}_N) d(c) + o(1). \tag{10}$$

Equation (10) suggests that for large enough $N$, the maximum coverage is only dependent on the *number* of the monitoring nodes allocated to each community. Also, the allocation is the same for all random instances so we can drop the dependence of $\boldsymbol{s}$ on $\mathbb{S}_N$. In right-hand side of Equation (10), the first term is the within-community (Lemma 4), and the second term is the between-community (Lemma 6) coverage.

In the analysis below, all the evaluations are for large enough $N$. Therefore, we drop the $\lim_{N \to \infty}$ for ease of notation. According to Equation (10) the between-community coverage is negligible, compared to the within-community coverage. This suggests that the maximum achievable coverage will be obtained by placing all the monitoring nodes in the largest community, with the largest value of $d(c)$, where the assumption on $I$, as given in the premise of the proposition, combined with Lemma 3 guarantee that such a selection is possible. Thus, we obtain

$$\mathbb{E}[\text{OPT}(\mathbb{S}_N, I, 0)] = Id(C) + o(1).$$

Next, we measure $\mathbb{E}[\text{OPT}^{\text{fair}}(.)]$, where in addition to optimization problem in Equation (10), the allocation is further restricted to satisfy all the fairness constraints.

$$\frac{s_c}{|\mathcal{N}_c|} d(c) + o(1) \geq W \quad \forall c \in \mathcal{C}, \tag{11}$$

in which, $o(1)$ is the term that compensates for the coverage of the nodes in other communities, and is small due to the regimes of $p_{cc'}^{\text{out}}$, $\forall c, c' \in \mathcal{C}$ and the budget $I$. At optimality and for the maximum value of $W$, we have

$$\left| s_c |\mathcal{N}_c|^{-1} d(c) - s_{c'} |\mathcal{N}_{c'}|^{-1} d(c') \right| \leq \delta \ \forall c, c' \in \mathcal{C}, \delta \leq \left| d(1) |\mathcal{N}_1|^{-1} - d(C) |\mathcal{N}_C|^{-1} \right|.$$

This holds because otherwise one can remove on node from the group with higher value of $s_c |\mathcal{N}_c|^{-1} d(c)$ to a group with less value and thus increase the normalized coverage of the worse-off

group and this contradicts the fact that $W$ is the maximum possible value. This suggests that in a fair solution, the normalized coverage is *almost* equal across different groups, given that $\lim_{N\to\infty} \delta = 0$. As a result, the monitoring nodes should be such that

$$W \leq \frac{s_c}{|\mathcal{N}_c|} d(c) + o(1) \leq W + \delta, \ \forall c \in \mathcal{C}.$$

From this, it follows that

$$W - o(1) \leq \frac{s_c}{|\mathcal{N}_c|} d(c) \leq W + o(1). \tag{12}$$

By assumption, there must be an integral $s_c$ that satisfies the above relation. Note that if we could relax the integrality assumption, $s_c = W|\mathcal{N}_c|d(c)^{-1}$. Due to the integrality constraint, and according to Equation (12), we set $s_c|\mathcal{N}_c|^{-1}d(c) = W + o(1)$, where the o(1) term is to account for the discretizing error, which results in $s_c = W|\mathcal{N}_c|d(c)^{-1} + O(1)$, where $O(1) \leq 1$ (As we can not make a higher error in rounding). Also, since $\sum_{c\in\mathcal{C}} s_c = I$, we can obtain the value of $W$ as

$$W = \frac{I}{\sum_{c\in\mathcal{C}} \frac{|\mathcal{N}_c|}{d(c)}} + o(1).$$

As a result

$$s_c = \frac{I}{\sum_{c\in\mathcal{C}} \frac{|\mathcal{N}_c|}{d(c)}} \frac{|\mathcal{N}_c|}{d(c)} + O(1) \quad \forall c \in \mathcal{C}.$$

We now define $\kappa := I \left( \sum_{c\in\mathcal{C}} \frac{|\mathcal{N}_c|}{d(c)} \right)^{-1}$ for a compact representation.

So far, we obtained the allocation of the monitoring nodes to satisfy the fairness constraints. This is enough to evaluate the coverage under the fairness constraints. Now, we can evaluate the PoF as defined by Equation (2).

$$\mathbb{E}[\mathrm{OPT}(\mathbb{S}_N, I, 0)] = I d(C)$$

$$\Rightarrow \quad -\frac{1}{\mathbb{E}[\mathrm{OPT}(\mathbb{S}_N, I, 0)]} = -\frac{1}{I\,d(C)}$$

$$\Rightarrow \quad -\frac{\mathbb{E}[\mathrm{OPT}^{\mathrm{fair}}(\mathbb{S}_N, I, 0)]}{\mathbb{E}[\mathrm{OPT}(\mathbb{S}_N, I, 0)]} = -\frac{\kappa \sum_{c\in\mathcal{C}} \frac{|\mathcal{N}_c|}{d(c)} d(c)}{I\,d(C)} - o(1)$$

$$\Rightarrow \quad 1 - \frac{\mathbb{E}[\mathrm{OPT}^{\mathrm{fair}}(\mathbb{S}_N, I, 0)]}{\mathbb{E}[\mathrm{OPT}(\mathbb{S}_N, I, 0)]} = 1 - \frac{\kappa \sum_{c\in\mathcal{C}} \frac{|\mathcal{N}_c|}{d(c)} d(c)}{I\,d(C)} - o(1)$$

$$\Rightarrow \quad \overline{\mathrm{PoF}}(I, 0) = 1 - \frac{\kappa \sum_{c\in\mathcal{C}} |\mathcal{N}_c|}{I\,d(C)} - o(1)$$

$$\Rightarrow \quad \overline{\mathrm{PoF}}(I, 0) = 1 - \frac{\sum_{c\in\mathcal{C}} |\mathcal{N}_c|}{\sum_{c\in\mathcal{C}} |\mathcal{N}_c|d(C)/d(c)} - o(1).$$

∎

### C.4 PoF in the Robust Case

*Proof of Proposition 2.* The idea of the proof is similar to Proposition 1, with the exception that the fair allocation of the monitoring nodes will be affected by the uncertainty. Consider $s$ to be the number of allocated monitoring nodes to each of the $C$ communities, i.e., $s_c = \sum_{n\in\mathcal{N}_c} x_n$. Using the result of lemma 4, and 6, we can measure the expected maximum coverage as

$$\mathbb{E}[\mathrm{OPT}(\mathbb{S}_N, I, J)] = (I - J)d(c) + o(1).$$

That is because, in the worst-case $J$ nodes fail, thus only $(I - J)$ nodes can cover the graph. Next, we measure $\mathbb{E}[\mathrm{OPT}^{\mathrm{fair}}(.)]$, where in addition to optimization problem in Equation (10), the allocation is further restricted to satisfy all the fairness constraints. Given that at most $J$ nodes may fail, we

need to ensure after fairness constraints are satisfied after the removal of $J$ nodes. We momentarily revisit the fairness constraint in the deterministic case.

$$\frac{s_c}{|\mathcal{N}_c|}d(c) + o(1) \geq W \quad \forall c \in \mathcal{C},$$

in which, $o(1)$ is the term that compensates for the coverage of the nodes in other communities, and is small due to the regimes of $p^{\text{out}}$, and the budget $I$. Under the uncertainty, we need to ensure that these constraints are satisfied even after $J$ nodes are removed. In other words

$$\frac{(s_c - J)}{|\mathcal{N}_c|}d(c) + o(1) \geq W \quad \forall c \in \mathcal{C}.$$

At optimality and for the maximum value of $W$, we have

$$\left| (s_c - J)|\mathcal{N}_c|^{-1}d(c) - (s_{c'} - J)|\mathcal{N}_{c'}|^{-1}d(c') \right| \leq \delta \ \forall c, c' \in \mathcal{C}, \delta \leq \left| d(1)|\mathcal{N}_1|^{-1} - d(C)|\mathcal{N}_C|^{-1} \right|.$$

This holds because otherwise one can remove on node from the group with higher value of $s_c|\mathcal{N}_c|^{-1}d(c)$ to a group with less value and thus increase the normalized coverage of the worse-off group and this contradicts the fact that $W$ is the maximum possible value.

This suggests that in a fair solution, the normalized coverage is *almost* equal across different groups, given that $\delta \to 0$, as $\mathcal{N}_c \to \infty, \forall c \in \mathcal{C}$. Following the proof of Proposition 1, the discretizing error can be handled by setting $(s_c - J)|\mathcal{N}_c|^{-1}d(c) = W + o(1)$, where the o(1) term is to account for the discretizing error. As a result

$$s_c = \frac{|\mathcal{N}_c|W}{d(c)} + J + O(1),$$

where $O(1) \leq 1$ (As we can not make a higher error in rounding). This suggests that a fair allocation is the one that places $J$ nodes in each community, regardless of the community size. The remaining monitors are allocated with respect to the relative size of the communities.

Summing over all $s_c$ and since $\sum_{c \in \mathcal{C}} s_c = I$ we obtain

$$W = \frac{(I - CJ)}{\sum_{c \in \mathcal{C}} \frac{|\mathcal{N}_c|}{d(c)}} + o(1).$$

As a result

$$s_c = \frac{(I - CJ)}{\sum_{c \in \mathcal{C}} \frac{|\mathcal{N}_c|}{d(c)}} \frac{|\mathcal{N}_c|}{d(c)} + J + O(1) \quad \forall c \in \mathcal{C}.$$

As defined in the premise of the proposition, $\eta = (I - CJ)\left(\sum_{c \in \mathcal{C}} \frac{|\mathcal{N}_c|}{d(c)}\right)^{-1}$.

So far, we obtained the allocation of the monitoring nodes, to satisfy the fairness constraints.

Now, we evaluate the coverage, i.e., objective value of Problem ($\overline{\mathcal{RC}}_{\text{fair}}$), under the obtained fair allocation. Since the fairness constraints are satisfied under all the scenarios, the worst-case scenario is the one that results in the maximum loss in the total coverage. This corresponds to the case that $J$ nodes from the largest community ($\mathcal{N}_C$) fail. As a result the expected coverage can be obtained by

$$\mathbb{E}[\text{OPT}^{\text{fair}}(\mathbb{S}_N, I, J)] = \sum_{c \in \mathcal{C}} \left( \eta \frac{|\mathcal{N}_c|}{d(c)}d(c) + Jd(c) + O(1)d(c) \right) - Jd(C).$$

Now, we can evaluate the PoF as defined by Equation (2).

$$
\begin{aligned}
\mathbb{E}[\text{OPT}(\mathbb{S}_N, I, J)] &= (I - J)d(C) \\
\Rightarrow \quad -\frac{1}{\mathbb{E}[\text{OPT}(\mathbb{S}_N, I, J)]} &= -\frac{1}{(I - J)d(C)} \\
\Rightarrow \quad -\frac{\mathbb{E}[\text{OPT}^{\text{fair}}(\mathbb{S}_N, I, J)]}{\mathbb{E}[\text{OPT}(\mathbb{S}_N, I, J)]} &= -\frac{\sum_{c \in \mathcal{C}} (\eta|\mathcal{N}_c| + Jd(c)) - Jd(C)}{(I - J)d(C)} - o(1) \\
\Rightarrow \quad 1 - \frac{\mathbb{E}[\text{OPT}^{\text{fair}}(\mathbb{S}_N, I, J)]}{\mathbb{E}[\text{OPT}(\mathbb{S}_N, I, J)]} &= 1 - \frac{\sum_{c \in \mathcal{C}} \eta|\mathcal{N}_c| + \sum_{c \in \mathcal{C}\backslash\{C\}} Jd(c)}{(I - J)d(C)} - o(1) \\
\Rightarrow \quad \overline{\text{PoF}}(I, J) &= 1 - \frac{\sum_{c \in \mathcal{C}} \eta|\mathcal{N}_c|}{(I - J)d(C)} - \frac{J\sum_{c \in \mathcal{C}\backslash\{C\}} d(c)}{(I - J)d(C)} - o(1).
\end{aligned}
$$

∎

# D  Supplemental Material: Proofs of Statements in Section 4

## D.1  Equivalent Reformulation as a Max-Min-Max Robust Optimization Problem

*Proof of Proposition 3.* Let $\bar{x}$ be feasible in Problem ($\mathcal{RC}_{\text{fair}}$). It follows that it is also feasible in Problem 3. For a fixed $\bar{\xi}$, we show that

$$\sum_{c \in \mathcal{C}} F_{\mathcal{G},c}(\bar{\boldsymbol{x}}, \bar{\boldsymbol{\xi}}) = \max_{\boldsymbol{y}} \quad \sum_{c \in \mathcal{C}} \sum_{n \in \mathcal{N}_c} \boldsymbol{y}_n$$

$$\text{s.t.} \quad \boldsymbol{y}_n \le \sum_{\nu \in \delta(n)} \bar{\boldsymbol{\xi}}_\nu \bar{\boldsymbol{x}}_\nu$$

$$\sum_{n \in \mathcal{C}} \boldsymbol{y}_n \ge W |\mathcal{N}_c|, \ \forall c \in \mathcal{C}$$

Since $\bar{x}$ is feasible in Problem ($\mathcal{RC}_{\text{fair}}$), it holds that

$$F_{\mathcal{G},c}(\bar{\boldsymbol{x}}, \bar{\boldsymbol{\xi}}) = \sum_{n \in \mathcal{N}_c} \boldsymbol{y}_n(\bar{\boldsymbol{x}}, \bar{\boldsymbol{\xi}})$$

$$= \sum_{n \in \mathcal{N}_c} \mathbb{I}\left( \sum_{\nu \in \delta(n)} \bar{\boldsymbol{\xi}}_\nu \bar{\boldsymbol{x}}_\nu \ge 1 \right)$$

$$\ge W|\mathcal{N}_c|$$

We define $\boldsymbol{y}_n^\star = \mathbb{I}\left( \sum_{\nu \in \delta(n)} \bar{\boldsymbol{\xi}}_\nu \bar{\boldsymbol{x}}_\nu \ge 1 \right)$ which is feasible in Problem (3). Since the choice of $\bar{\boldsymbol{\xi}}$ was arbitrary, we showed that given a solution to Problem ($\mathcal{RC}_{\text{fair}}$), we can always construct a feasible solution to Problem (3), thus the objective value of the latter is at least as high.

We now prove the contrary, i.e., given a solution to Problem (3), we will construct a solution to Problem ($\mathcal{RC}_{\text{fair}}$). Consider $\bar{x}$ to be an optimal solution to Problem ($\mathcal{RC}_{\text{fair}}$). Suppose there exists $\bar{\boldsymbol{\xi}} \in \Xi$ such that

$$F_{\mathcal{G},c}(\bar{\boldsymbol{x}}, \bar{\boldsymbol{\xi}}) \quad < \quad |\mathcal{N}_c|W$$

$$\Rightarrow \sum_{n \in \mathcal{N}_c} \mathbb{I}\left( \sum_{\nu \in \delta(n)} \bar{\boldsymbol{\xi}}_\nu \bar{\boldsymbol{x}}_\nu \ge 1 \right) \quad < \quad |\mathcal{N}_c|W.$$

However, since $\bar{x}$ is feasible in Problem ($\mathcal{RC}_{\text{fair}}$), we have that

$$\forall \tilde{\boldsymbol{\xi}} \in \Xi, \ \exists \boldsymbol{y}_n : \quad \boldsymbol{y}_n \le \sum_{\nu \in \delta(n)} \tilde{\boldsymbol{\xi}}_\nu \bar{\boldsymbol{x}}_\nu$$

$$\sum_{n \in \mathcal{N}_c} \boldsymbol{y}_n \ge |\mathcal{N}_c|W.$$

By construction, $\boldsymbol{y}_n \le \mathbb{I}\left( \sum_{\nu \in \delta(n)} \tilde{\boldsymbol{\xi}}_\nu \bar{\boldsymbol{x}}_\nu \ge 1 \right), \ \forall n \in \mathcal{N}$. Thus

$$\sum_{c \in \mathcal{C}} \sum_{n \in \mathcal{N}_c} \mathbb{I}\left( \sum_{\nu \in \delta(n)} \tilde{\boldsymbol{\xi}}_\nu \bar{\boldsymbol{x}}_\nu \ge 1 \right) \quad \ge \quad \sum_{c \in \mathcal{C}} \sum_{n \in \mathcal{N}_c} \boldsymbol{y}_n$$

$$\ge \quad |\mathcal{N}_c|W.$$

According to the above result, we showed that the optimal objective value of Problem ($\mathcal{RC}_{\text{fair}}$) is at least as high as that of Problem (3). This completes the proof. ∎

## D.2 Exact MILP Formulation of the K-Adaptability Problem

In order to derive the equivalent MILP in Theorem 1, we start by a variant of the $K$-adaptability Problem (4), in which we move the constraints of the inner maximization problem to the definition of the uncertainty set in the spirit of [31]. Next, we prove, via Proposition 4, that by relaxing the integrality constraint on the uncertain parameters $\boldsymbol{\xi}$, the problem remains unchanged, and this is the key result that enables us to provide an *equivalent* MILP reformulation for Problem (4).

We replace $\Xi$ with a collection of uncertainty sets parameterized by vectors $\boldsymbol{\ell} \in \mathcal{L}$ as in [31]. Specifically, it follows from Proposition 2 in [31] that Problem (4) is equivalent to

$$\max \quad \min_{\boldsymbol{\ell} \in \mathcal{L}} \quad \min_{\boldsymbol{\xi} \in \Xi(\boldsymbol{x}, \boldsymbol{y}, \boldsymbol{\ell})} \quad \max_{\substack{k \in \mathcal{K}: \\ \boldsymbol{\ell}_k = 0}} \sum_{n \in \mathcal{N}} \boldsymbol{y}_n^k \tag{13}$$
$$\text{s.t.} \quad \boldsymbol{x} \in \mathcal{X}, \ \boldsymbol{y}^1, \ldots, \boldsymbol{y}^K \in \mathcal{Y},$$

where $\Xi(\boldsymbol{x}, \boldsymbol{y}, \boldsymbol{\ell})$ is defined through

$$\Xi(\boldsymbol{x}, \boldsymbol{y}, \boldsymbol{\ell}) := \left\{ \boldsymbol{\xi} \in \Xi : \begin{array}{ll} \boldsymbol{y}_{\boldsymbol{\ell}_k}^k > \displaystyle\sum_{\nu \in \delta(\boldsymbol{\ell}_k)} \boldsymbol{\xi}_\nu \boldsymbol{x}_\nu, & \forall k \in \mathcal{K} : \boldsymbol{\ell}_k > 0 \\[4mm] \boldsymbol{y}_n^k \leq \displaystyle\sum_{\nu \in \delta(n)} \boldsymbol{\xi}_\nu \boldsymbol{x}_\nu \ \ \forall n \in \mathcal{N}, \ \forall k \in \mathcal{K} : \boldsymbol{\ell}_k = 0 \end{array} \right\},$$

and, with a slight abuse of notation, we use $\boldsymbol{y} := \{\boldsymbol{y}^1, \ldots, \boldsymbol{y}^K\}$. The vector $\boldsymbol{\ell} \in \mathcal{L}$ encodes which of the $K$ candidate covering schemes are feasible. By introducing $\boldsymbol{\ell}$, the constraints of the inner maximization problem are absorbed in the parameterized uncertainty sets $\Xi(\boldsymbol{x}, \boldsymbol{y}, \boldsymbol{\ell})$, and in the inner-most maximization problem, any covering scheme can be chosen for which $\boldsymbol{\ell}_k = 0$.

Note that, for any fixed $\boldsymbol{x} \in \mathcal{X}$, $\boldsymbol{y} \in \mathcal{Y}^K$, and $\boldsymbol{\ell} \in \mathcal{L}$, the strict inequalities in $\Xi(\boldsymbol{x}, \boldsymbol{y}, \boldsymbol{\ell})$ can be converted to (loose) inequalities as in

$$\Xi(\boldsymbol{x}, \boldsymbol{y}, \boldsymbol{\ell}) = \left\{ \boldsymbol{\xi} \in \Xi : \begin{array}{ll} \boldsymbol{y}_{\boldsymbol{\ell}_k}^k \geq \displaystyle\sum_{\nu \in \delta(\boldsymbol{\ell}_k)} \boldsymbol{\xi}_\nu \boldsymbol{x}_\nu + 1, & \forall k \in \mathcal{K} : \boldsymbol{\ell}_k > 0 \\[4mm] \boldsymbol{y}_n^k \leq \displaystyle\sum_{\nu \in \delta(n)} \boldsymbol{\xi}_\nu \boldsymbol{x}_\nu \ \ \forall n \in \mathcal{N}, \ \forall k \in \mathcal{K} : \boldsymbol{\ell}_k = 0 \end{array} \right\}.$$

This idea was previously leveraged in [39]. It follows naturally since all decision variables and uncertain parameters are binary. Next, we show that we can obtain an equivalent problem by relaxing the integrality constraint on the set $\Xi$ in the definition of $\Xi(\boldsymbol{x}, \boldsymbol{y}, \boldsymbol{l})$. Consider the following problem

$$\max \quad \min_{\boldsymbol{\ell} \in \mathcal{L}} \quad \min_{\boldsymbol{\xi} \in \overline{\Xi}(\boldsymbol{x}, \boldsymbol{y}, \boldsymbol{\ell})} \quad \max_{\substack{k \in \mathcal{K}: \\ \boldsymbol{\ell}_k = 0}} \sum_{n \in \mathcal{N}} \boldsymbol{y}_n^k \tag{14}$$
$$\text{s.t.} \quad \boldsymbol{x} \in \mathcal{X}, \ \boldsymbol{y} \in \mathcal{Y}^K,$$

where the uncertainty set is obtained by relaxing the integrality constraints on $\boldsymbol{\xi}$, i.e.,

$$\overline{\Xi}(\boldsymbol{x}, \boldsymbol{y}, \boldsymbol{\ell}) = \left\{ \boldsymbol{\xi} \in \mathcal{T} : \begin{array}{ll} \boldsymbol{y}_{\boldsymbol{\ell}_k}^k \geq \displaystyle\sum_{\nu \in \delta(\boldsymbol{\ell}_k)} \boldsymbol{\xi}_\nu \boldsymbol{x}_\nu + 1, & \forall k \in \mathcal{K} : \boldsymbol{\ell}_k > 0 \\[4mm] \boldsymbol{y}_n^k \leq \displaystyle\sum_{\nu \in \delta(n)} \boldsymbol{\xi}_\nu \boldsymbol{x}_\nu \ \ \forall n \in \mathcal{N}, \ \forall k \in \mathcal{K} : \boldsymbol{\ell}_k = 0 \end{array} \right\}.$$

**Proposition 4.** *Under Assumption 3, Problems (13) and (14) are equivalent.*

**Proof.** Let $\boldsymbol{x} \in \mathcal{X}$, $\boldsymbol{y} \in \mathcal{Y}^K$, and $\boldsymbol{\ell} \in \mathcal{L}$. It suffices to show that

$$\min_{\boldsymbol{\xi} \in \Xi(\boldsymbol{x}, \boldsymbol{y}, \boldsymbol{\ell})} \quad \max_{\substack{k \in \mathcal{K}: \\ \boldsymbol{\ell}_k = 0}} \sum_{n \in \mathcal{N}} \boldsymbol{y}_n^k \qquad \text{and} \qquad \min_{\boldsymbol{\xi} \in \overline{\Xi}(\boldsymbol{x}, \boldsymbol{y}, \boldsymbol{\ell})} \quad \max_{\substack{k \in \mathcal{K}: \\ \boldsymbol{\ell}_k = 0}} \sum_{n \in \mathcal{N}} \boldsymbol{y}_n^k$$

are equivalent. Observe that the these problems have the same objective function. Thus, the two problems have the same optimal objective value if and only if they are either both feasible or both infeasible. As a result, it suffices to show that $\Xi(\boldsymbol{x}, \boldsymbol{y}, \boldsymbol{\ell})$ is empty if and only if $\overline{\overline{\Xi}}(\boldsymbol{x}, \boldsymbol{y}, \boldsymbol{\ell})$ is empty. Naturally, if $\overline{\overline{\Xi}}(\boldsymbol{x}, \boldsymbol{y}, \boldsymbol{\ell}) = \emptyset$ then $\Xi(\boldsymbol{x}, \boldsymbol{y}, \boldsymbol{\ell}) = \emptyset$ since $\mathcal{T}$ is the linear programming relaxation of $\Xi$. Thus, it suffices to show that the converse also holds, i.e., that if $\overline{\overline{\Xi}}(\boldsymbol{x}, \boldsymbol{y}, \boldsymbol{\ell}) \neq \emptyset$, then also $\Xi(\boldsymbol{x}, \boldsymbol{y}, \boldsymbol{\ell}) \neq \emptyset$.

To this end, suppose that $\overline{\overline{\Xi}}(\boldsymbol{x}, \boldsymbol{y}, \boldsymbol{\ell}) \neq \emptyset$ and let $\tilde{\boldsymbol{\xi}} \in \overline{\Xi}(\boldsymbol{x}, \boldsymbol{y}, \boldsymbol{\ell})$. Then, $\tilde{\boldsymbol{\xi}}$ is such that

$$
\begin{aligned}
\tilde{\boldsymbol{\xi}} &\in \mathcal{T}, \\
\boldsymbol{y}_{\boldsymbol{\ell}_k}^k &\geq \sum_{\nu \in \delta(\boldsymbol{\ell}_k)} \tilde{\boldsymbol{\xi}}_\nu \boldsymbol{x}_\nu + 1 \quad \forall k \in \mathcal{K} : \boldsymbol{\ell}_k > 0, \\
\boldsymbol{y}_n^k &\leq \sum_{\nu \in \delta(n)} \tilde{\boldsymbol{\xi}}_\nu \boldsymbol{x}_\nu \quad \forall n \in \mathcal{N}, \ \forall k \in \mathcal{K} : \boldsymbol{\ell}_k = 0.
\end{aligned}
\tag{15}
$$

Next, define $\hat{\boldsymbol{\xi}}_n := \lceil \tilde{\boldsymbol{\xi}}_n \rceil \ \forall n \in \mathcal{N}$. We show that $\hat{\boldsymbol{\xi}} \in \Xi(\boldsymbol{x}, \boldsymbol{y}, \boldsymbol{\ell})$. First, note that $\hat{\boldsymbol{\xi}} \geq \tilde{\boldsymbol{\xi}}$ and by Assumption 3, it follows that $\hat{\boldsymbol{\xi}} \in \mathcal{T}$. Moreover, by construction, $\hat{\boldsymbol{\xi}} \in \{0, 1\}^N$. Thus, it follows that $\hat{\boldsymbol{\xi}} \in \Xi$. Next, we show that the constructed solution $\hat{\boldsymbol{\xi}}$ also satisfies the remaining constraints in $\Xi(\boldsymbol{x}, \boldsymbol{y}, \boldsymbol{\ell})$. Fix $k \in \mathcal{K}$ such that $\boldsymbol{\ell}_k > 0$. Then, from (15) it holds that

$$
\begin{aligned}
& \boldsymbol{y}_{\boldsymbol{\ell}_k}^k \geq \sum_{\nu \in \delta(\boldsymbol{\ell}_k)} \tilde{\boldsymbol{\xi}}_\nu \boldsymbol{x}_\nu + 1 \\
\Rightarrow \quad & \boldsymbol{y}_{\boldsymbol{\ell}_k}^k = 1 \text{ and } \tilde{\boldsymbol{\xi}}_\nu \boldsymbol{x}_\nu = 0 \quad \forall \nu \in \delta(\boldsymbol{\ell}_k) \\
\Rightarrow \quad & \boldsymbol{y}_{\boldsymbol{\ell}_k}^k = 1 \text{ and } \tilde{\boldsymbol{\xi}}_\nu = 0 \quad \forall \nu \in \delta(\boldsymbol{\ell}_k) : \boldsymbol{x}_\nu = 1 \\
\Rightarrow \quad & \boldsymbol{y}_{\boldsymbol{\ell}_k}^k = 1 \text{ and } \hat{\boldsymbol{\xi}}_\nu = 0 \quad \forall \nu \in \delta(\boldsymbol{\ell}_k) : \boldsymbol{x}_\nu = 1 \\
\Rightarrow \quad & \boldsymbol{y}_{\boldsymbol{\ell}_k}^k \geq \sum_{\nu \in \delta(\boldsymbol{\ell}_k)} \hat{\boldsymbol{\xi}}_\nu \boldsymbol{x}_\nu + 1,
\end{aligned}
$$

where the first and second implication follow since $\boldsymbol{y}$ and $\boldsymbol{x}$ are binary, respectively, and the third implication holds by definition of $\hat{\boldsymbol{\xi}}$,

Next, fix $k \in \mathcal{K}$ such that $\boldsymbol{\ell}_k = 0$. Then, (15) yields

$$
\begin{aligned}
& \boldsymbol{y}_n^k \leq \sum_{\nu \in \delta(n)} \tilde{\boldsymbol{\xi}}_\nu \boldsymbol{x}_\nu \quad \forall n \in \mathcal{N} \\
\Rightarrow \quad & \boldsymbol{y}_n^k \leq \sum_{\nu \in \delta(n)} \hat{\boldsymbol{\xi}}_\nu \boldsymbol{x}_\nu \quad \forall n \in \mathcal{N},
\end{aligned}
$$

which follows by definition of $\hat{\boldsymbol{\xi}}$. We have thus constructed $\hat{\boldsymbol{\xi}} \in \Xi(\boldsymbol{x}, \boldsymbol{y}, \boldsymbol{\ell})$ and therefore conclude that $\Xi(\boldsymbol{x}, \boldsymbol{y}, \boldsymbol{\ell}) \neq \emptyset$. Since the choice of $\boldsymbol{x} \in \mathcal{X}$, $\boldsymbol{y} \in \mathcal{Y}^K$, and $\boldsymbol{\ell} \in \mathcal{L}$ was arbitrary, the claim follows. $\blacksquare$

Proposition 4 is key to leverage existing literature to reformulate Problem (4) as an MILP. The reformulation is based on [31, 39].

*Proof of Theorem 1.* Note that the objective function of the Problem (13) is identical to

$$
\min_{\boldsymbol{\ell} \in \mathcal{L}} \min_{\boldsymbol{\xi} \in \overline{\overline{\Xi}}(\boldsymbol{x}, \boldsymbol{y}, \boldsymbol{\ell})} \left[ \max_{\boldsymbol{\lambda} \in \Delta_K(\boldsymbol{\ell})} \sum_{k \in \mathcal{K}} \boldsymbol{\lambda}_k \sum_{n \in \mathcal{N}} \boldsymbol{y}_n^k \right],
$$

where $\Delta_K(\boldsymbol{\ell}) := \{ \boldsymbol{\lambda} \in \mathbb{R}_+^K : \mathbf{e}^\top \boldsymbol{\lambda} = 1, \ \boldsymbol{\lambda}_k = 0 \ \forall k \in \mathcal{K} : \boldsymbol{\ell}_k \neq 0 \}$. We define $\partial \mathcal{L} := \{ \boldsymbol{\ell} \in \mathcal{L} : \boldsymbol{\ell} \not> \boldsymbol{0} \}$, and $\mathcal{L}_+ := \{ \boldsymbol{\ell} \in \mathcal{L} : \boldsymbol{\ell} > \boldsymbol{0} \}$. We remark that $\Delta_K(\boldsymbol{\ell}) = \emptyset$ if and only if $\boldsymbol{\ell} > \boldsymbol{0}$. If $\overline{\overline{\Xi}}(\boldsymbol{x}, \boldsymbol{y}, \boldsymbol{\ell}) = \emptyset$ for all $\boldsymbol{\ell} \in \mathcal{L}_+$, then the problem is equivalent to

$$
\min_{\boldsymbol{\ell} \in \partial \mathcal{L}} \min_{\boldsymbol{\xi} \in \overline{\overline{\Xi}}(\boldsymbol{x}, \boldsymbol{y}, \boldsymbol{\ell})} \left[ \max_{\boldsymbol{\lambda} \in \Delta_K(\boldsymbol{\ell})} \sum_{k \in \mathcal{K}} \boldsymbol{\lambda}_k \sum_{n \in \mathcal{N}} \boldsymbol{y}_n^k \right].
$$

By applying the classical min-max theorem, we obtain

$$\min_{\boldsymbol{\ell}\in\partial\mathcal{L}} \quad \max_{\boldsymbol{\lambda}\in\Delta_K(\boldsymbol{\ell})} \quad \min_{\boldsymbol{\xi}\in\overline{\Xi}(\boldsymbol{x},\boldsymbol{y},\boldsymbol{\ell})} \sum_{k\in\mathcal{K}}\boldsymbol{\lambda}_k \sum_{n\in\mathcal{N}} \boldsymbol{y}_n^k.$$

This problem is also equivalent to

$$\max_{\boldsymbol{\lambda}(\boldsymbol{\ell})\in\Delta_K(\boldsymbol{\ell})} \quad \min_{\boldsymbol{\ell}\in\partial\mathcal{L}} \quad \min_{\boldsymbol{\xi}\in\overline{\Xi}(\boldsymbol{x},\boldsymbol{y},\boldsymbol{\ell})} \sum_{k\in\mathcal{K}}\boldsymbol{\lambda}_k(\boldsymbol{\ell}) \sum_{n\in\mathcal{N}} \boldsymbol{y}_n^k.$$

If on the other hand $\overline{\Xi}(\boldsymbol{x},\boldsymbol{y},\boldsymbol{\ell}) \neq \emptyset$ for some $\boldsymbol{\ell}\in\mathcal{L}_+$, the objective of Problem (13) evaluates to $-\infty$.

Using the above results, we can write Problem (13) in epigraph form as

$$
\begin{aligned}
\max \quad & \tau \\
\text{s.t.} \quad & \boldsymbol{x}\in\mathcal{X},\ \boldsymbol{y}\in\mathcal{Y}^K,\ \tau\in\mathbb{R},\ \boldsymbol{\lambda}(\boldsymbol{\ell})\in\Delta_K(\boldsymbol{\ell}),\ \boldsymbol{\ell}\in\partial\mathcal{L} \\
& \tau \leq \sum_{k\in\mathcal{K}}\boldsymbol{\lambda}_k(\boldsymbol{\ell})\sum_{n\in\mathcal{N}}\boldsymbol{y}_n^k \quad \forall\boldsymbol{\ell}\in\partial\mathcal{L}: \Xi(\boldsymbol{x},\boldsymbol{y},\boldsymbol{\ell})\neq\emptyset \\
& \overline{\Xi}(\boldsymbol{x},\boldsymbol{y},\boldsymbol{\ell})=\emptyset \quad \forall\boldsymbol{\ell}\in\mathcal{L}_+.
\end{aligned}
\tag{16}
$$

We begin by reformulating the semi-infinite constraint associated with $\boldsymbol{\ell}\in\partial\mathcal{L}$ in Problem (16). To this end, fix $\boldsymbol{\ell}\in\partial\mathcal{L}$ and consider the linear program

$$
\begin{aligned}
\min \quad & 0 \\
\text{s.t.} \quad & 0 \leq \boldsymbol{\xi}_n \leq 1 \ \ \forall n\in\mathcal{N} \\
& \boldsymbol{A}^\top\boldsymbol{\xi} \geq \boldsymbol{b} \\
& \boldsymbol{y}_{\boldsymbol{\ell}_k}^k \geq \sum_{\nu\in\delta(\boldsymbol{\ell}_k)}\boldsymbol{\xi}_\nu\boldsymbol{x}_\nu+1 \quad \forall k\in\mathcal{K}: \boldsymbol{\ell}_k>0 \\
& \boldsymbol{y}_n^k \leq \sum_{\nu\in\delta(n)}\boldsymbol{\xi}_\nu\boldsymbol{x}_\nu \quad \forall n\in\mathcal{N},\ \forall k\in\mathcal{K}: \boldsymbol{\ell}_k=0,
\end{aligned}
$$

whose dual reads

$$
\begin{aligned}
\max \quad & -\mathbf{e}^\top\boldsymbol{\theta}(\boldsymbol{\ell})+\boldsymbol{b}^\top\boldsymbol{\alpha}(\boldsymbol{\ell})-\sum_{\substack{k\in\mathcal{K}\\\boldsymbol{\ell}_k\neq0}}\left(\boldsymbol{y}_{\boldsymbol{\ell}_k}^k-1\right)\boldsymbol{\nu}_k(\boldsymbol{\ell})+\sum_{\substack{k\in\mathcal{K}\\\boldsymbol{\ell}_k=0}}\sum_{n\in\mathcal{N}}\boldsymbol{y}_n^k\boldsymbol{\beta}_n^k(\boldsymbol{\ell}) \\
\text{s.t.} \quad & \boldsymbol{\theta}(\boldsymbol{\ell})\in\mathbb{R}_+^N,\ \boldsymbol{\alpha}(\boldsymbol{\ell})\in\mathbb{R}_+^R,\ \boldsymbol{\beta}^k(\boldsymbol{\ell})\in\mathbb{R}_+^N,\ \forall k\in\mathcal{K},\ \boldsymbol{\nu}(\boldsymbol{\ell})\in\mathbb{R}_+^K \\
& \boldsymbol{\theta}_n(\boldsymbol{\ell}) \leq \boldsymbol{A}^\top\boldsymbol{\alpha}(\boldsymbol{\ell})+\sum_{\substack{k\in\mathcal{K}\\\boldsymbol{\ell}_k\neq0}}\sum_{\nu\in\delta(\boldsymbol{\ell}_k)}\boldsymbol{x}_\nu\boldsymbol{\nu}_k(\boldsymbol{\ell})-\sum_{\substack{k\in\mathcal{K}\\\boldsymbol{\ell}_k=0}}\sum_{\nu\in\delta(n)}\boldsymbol{x}_\nu\boldsymbol{\beta}_n^k(\boldsymbol{\ell}) \quad \forall n\in\mathcal{N}.
\end{aligned}
$$

In Problem (16) the constraint associated with each $\boldsymbol{\ell}\in\partial\mathcal{L}$ is satisfied if and only if the objective value of the above dual problem is greater than $\tau-\sum_{k\in\mathcal{K}}\boldsymbol{\lambda}_k(\boldsymbol{\ell})\sum_{n\in\mathcal{N}}\boldsymbol{y}_n^k$. This follows since the dual is always feasible. Therefore, either the dual is unbounded in which case the primal is infeasible, i.e., $\Xi(\boldsymbol{x},\boldsymbol{y},\boldsymbol{\ell})=\emptyset$, and the constraint is trivial. Else, by strong duality, the primal and dual must have the same objective value (zero). As a result, the constraints in Problem (16) associated with each $\boldsymbol{\ell}\in\partial\mathcal{L}$ can be written as

$$
\tau\leq-\mathbf{e}^\top\boldsymbol{\theta}(\boldsymbol{\ell})+\boldsymbol{b}^\top\boldsymbol{\alpha}(\boldsymbol{\ell})-\sum_{\substack{k\in\mathcal{K}\\\boldsymbol{\ell}_k\neq0}}\left(\boldsymbol{y}_{\boldsymbol{\ell}_k}^k-1\right)\boldsymbol{\nu}_k(\boldsymbol{\ell})+\sum_{\substack{k\in\mathcal{K}\\\boldsymbol{\ell}_k=0}}\sum_{n\in\mathcal{N}}\boldsymbol{y}_n^k\boldsymbol{\beta}_n^k(\boldsymbol{\ell})+\sum_{k\in\mathcal{K}}\boldsymbol{\lambda}_k(\boldsymbol{\ell})\sum_{n\in\mathcal{N}}\boldsymbol{y}_n^k
$$

$$
\boldsymbol{\theta}_n(\boldsymbol{\ell}) \leq \boldsymbol{A}^\top\boldsymbol{\alpha}(\boldsymbol{\ell})+\sum_{\substack{k\in\mathcal{K}\\\boldsymbol{\ell}_k\neq0}}\sum_{\nu\in\delta(\boldsymbol{\ell}_k)}\boldsymbol{x}_\nu\boldsymbol{\nu}_k(\boldsymbol{\ell})-\sum_{\substack{k\in\mathcal{K}\\\boldsymbol{\ell}_k=0}}\sum_{\nu\in\delta(n)}\boldsymbol{x}_\nu\boldsymbol{\beta}_n^k(\boldsymbol{\ell}) \quad \forall n\in\mathcal{N}.
$$

Finally, the last constraint in Problem ([16](#)) is satisfied if the linear program

$$
\begin{aligned}
\min \quad & 0 \\
\text{s.t.} \quad & 0 \;\leq\; \boldsymbol{\xi}_n \leq 1 \quad \forall n \in \mathcal{N} \\
& \boldsymbol{A}\boldsymbol{\xi} \;\geq\; \boldsymbol{b} \\
& \boldsymbol{y}_{\boldsymbol{\ell}_k}^k \;\geq\; \sum_{\nu \in \delta(\boldsymbol{\ell}_k)} \boldsymbol{\xi}_\nu \boldsymbol{x}_\nu + 1 \quad \forall k \in \mathcal{K} \,:\, \boldsymbol{\ell}_k \neq 0
\end{aligned}
$$

is infeasible. Using strong duality, this occurs if the dual problem

$$
\begin{aligned}
\max \quad & -\mathbf{e}^\top \boldsymbol{\theta}(l) + \boldsymbol{\alpha}(\boldsymbol{\ell})^\top \boldsymbol{b} - \sum_{\substack{k \in \mathcal{K} \\ \boldsymbol{\ell}_k \neq 0}} \left( \boldsymbol{y}_{\boldsymbol{\ell}_k}^k - 1 \right) \boldsymbol{\nu}_k(\boldsymbol{\ell}) \\
\text{s.t.} \quad & \boldsymbol{\theta}(\boldsymbol{\ell}) \in \mathbb{R}_+^N, \; \boldsymbol{\alpha}(\boldsymbol{\ell}) \in \mathbb{R}_+^R, \; \boldsymbol{\nu}(\boldsymbol{\ell}) \in \mathbb{R}_+^K \\
& \boldsymbol{\theta}_n(\boldsymbol{\ell}) \;\leq\; \boldsymbol{A}^\top \boldsymbol{\alpha}(\boldsymbol{\ell}) + \sum_{\substack{k \in \mathcal{K} \\ \boldsymbol{\ell}_k \neq 0}} \sum_{\nu \in \delta(\boldsymbol{\ell}_k)} \boldsymbol{x}_\nu \boldsymbol{\nu}_k(\boldsymbol{\ell}) \quad \forall n \in \mathcal{N}
\end{aligned}
$$

is unbounded. Since the feasible region of the dual problem constitutes a cone, the dual problem is unbounded if and only if there is a feasible solution with an objective value of 1 or more. ∎

## E    Supplemental Material: Bender's Decomposition

We do not detail all the steps of the Bender's decomposition algorithm. We merely provide the initial relaxed master problem and the subproblems used to generate the cuts. We refer the reader to e.g., [16] for more details.

**Relaxed Master Problem.** Initially, the relaxed master problem only involves the binary variables of the Problem ([5](#)) and is expressible as

$$
\max \; \left\{ \tau \,:\, \tau \in \mathbb{R}, \; \boldsymbol{x} \in \mathcal{X}, \; \boldsymbol{y}^1, \dots, \boldsymbol{y}^K \in \mathcal{Y} \right\}.
$$

**Subproblems.** As discussed in Section [4](#), Problem ([5](#)) decomposes by $\boldsymbol{\ell}$. Depending on the index $\boldsymbol{\ell}$ of the subproblem, there are two types of subproblems to consider. If $\boldsymbol{\ell} \in \mathcal{L}_0$, the subproblem is given by

$$
\begin{aligned}
\min \quad & 0 \\
\text{s.t.} \quad & \boldsymbol{\theta}(\boldsymbol{\ell}), \; \boldsymbol{\beta}^k(\boldsymbol{\ell}) \in \mathbb{R}_+^N, \; \boldsymbol{\alpha}(\boldsymbol{\ell}) \in \mathbb{R}_+^R, \; \boldsymbol{\nu}(\boldsymbol{\ell}) \in \mathbb{R}_+^K, \; \boldsymbol{\lambda}(\boldsymbol{\ell}) \in \Delta_K(\boldsymbol{\ell}) \\
& \tau \;\leq\; -\mathbf{e}^\top \boldsymbol{\theta}(\boldsymbol{\ell}) + \boldsymbol{b}^\top \boldsymbol{\alpha}(\boldsymbol{\ell}) - \sum_{\substack{k \in \mathcal{K}: \\ \boldsymbol{\ell}_k \neq 0}} \left( \boldsymbol{y}_{\boldsymbol{\ell}_k}^k - 1 \right) \boldsymbol{\nu}_k(\boldsymbol{\ell}) + \dots \\
& \dots + \sum_{\substack{k \in \mathcal{K}: \\ \boldsymbol{\ell}_k = 0}} \sum_{n \in \mathcal{N}} \boldsymbol{y}_n^k \boldsymbol{\beta}_n^k(\boldsymbol{\ell}) + \sum_{k \in \mathcal{K}} \boldsymbol{\lambda}_k(\boldsymbol{\ell}) \sum_{n \in \mathcal{N}} \boldsymbol{y}_n^k \qquad (\mathcal{Z}_0(\boldsymbol{\ell}))
\end{aligned}
$$

$$
\boldsymbol{\theta}_n(\boldsymbol{\ell}) \;\leq\; \boldsymbol{A}^\top \boldsymbol{\alpha}(\boldsymbol{\ell}) + \sum_{\substack{k \in \mathcal{K} \\ \boldsymbol{\ell}_k \neq 0}} \sum_{\nu \in \delta(l_k)} \boldsymbol{x}_\nu \boldsymbol{\nu}_k(\boldsymbol{\ell}) - \sum_{\substack{k \in \mathcal{K} \\ \boldsymbol{\ell}_k = 0}} \sum_{\nu \in \delta(n)} \boldsymbol{x}_\nu \boldsymbol{\beta}_n^k(\boldsymbol{\ell}) \;\; \forall n \in \mathcal{N}.
$$

In a similar fashion, we define the subproblem associated with $\boldsymbol{\ell} \in \mathcal{L}_+$, given by

$$
\begin{aligned}
\min \quad & 0 \\
\text{s.t.} \quad & \boldsymbol{\theta}(\boldsymbol{\ell}) \in \mathbb{R}_+^N, \; \boldsymbol{\alpha}(\boldsymbol{l}) \in \mathbb{R}_+^R, \; \boldsymbol{\nu}(\boldsymbol{l}) \in \mathbb{R}_+^K \\
& 1 \leq -\mathbf{e}^\top \boldsymbol{\theta}(\boldsymbol{l}) + \boldsymbol{b}^\top \boldsymbol{\alpha}(\boldsymbol{\ell}) - \sum_{\substack{k \in \mathcal{K} \\ \boldsymbol{\ell}_k \neq 0}} \left( y_{\boldsymbol{\ell}_k}^k - 1 \right) \boldsymbol{\nu}_k(\boldsymbol{\ell}) \\
& \boldsymbol{\theta}_n(\boldsymbol{\ell}) \leq \boldsymbol{A}^\top \boldsymbol{\alpha}(\boldsymbol{\ell}) + \sum_{\substack{k \in \mathcal{K} \\ \boldsymbol{\ell}_k \neq 0}} \sum_{\nu \in \delta(\boldsymbol{\ell}_k)} \boldsymbol{x}_\nu \boldsymbol{\nu}_k(\boldsymbol{\ell}) \quad \forall n \in \mathcal{N}.
\end{aligned}
\qquad (\mathcal{Z}_+(\boldsymbol{\ell}))
$$