[Reviews · NeurIPS 2019]

Reviewer 1



This paper addresses the problem of enforcing fairness in robust graph covering problems. The goal is to choose a subset of nodes in a graph to maximize the worst-case coverage, while maintaining that all protected groups receive some minimum amount of coverage (subsets of nodes make up protected groups). This paper deals additional uncertainty in the 'availability' of the chosen nodes: if a node is unavailable, then even if chosen, it will not cover its neighbors. They do not assume anything about the distribution of unavailable nodes. Originality: This paper's novelty comes from adding fairness constraints to robust graph covering. To my knowledge, this is the first work enforcing this particular minimum group coverage constraint in the context of graph covering. Quality: The quality of the experiments is hard for me to assess with certain details missing in the methodology (see Clarity and Improvements). This is the main issue holding me back from a higher score. The theoretical optimization problem setup seems solid, though Theorem 1 was hard to parse (see Clarity). The theoretical results regarding the price of fairness in Section 3 seem to be good justification for the paper's study of SBM networks. Clarity: The paper is organized well and presents the robust graph covering problem and price of fairness nicely in Sections 1-3. However, I have quite a few concerns with the clarity of the experimental results. There were many important experimental details that I could not find in either the main paper or the appendix (see Improvements). This hurts both the interpretability and reproducibility of the empirical results. Also, Theorem 1 is hard to parse. Can the authors summarize in words what the terms (or groups of terms) in the mixed-integer bilinear program generally represent? Significance: The proposed fairness metric of minimum group coverage seems to make sense in the practical settings that the authors cite of suicide prevention and disaster risk management. Thus, the setup and solution the authors propose seems to have practical significance.

Reviewer 2



The paper titled “Exploring Algorithmic Fairness in Robust Graph Covering Problems” deals with the shortcomings of traditional and even state-of-the-art graph covering solutions that result in biased node coverage. The implication of which in the real world could be the marginalization of some minority group, negligence towards a social sector etc. Therefore, even though the paper is theoretically sound and mature, it has some excellent real world applications. This is discussed in precise details later in the review. Briefly, the authors consider a network or graph of nodes/vertices where each node belongs to some group (race, gender etc.). The problem is to select a subset of the nodes that the authors call “monitors” (since they encompass neighboring nodes) along with the possibility that a fraction of them will fail to exist and thus the coverage by the failing nodes will be diminished. The success case is when a node is selected to be a non-failing monitor then that node’s neighbors would be covered by that node. The final objective is to maximize the coverage in the worst case scenario subject to a given fairness fraction (which is fixed by the authors) across the groups of nodes. If one considers the nodes to be people and neighboring nodes of a particular node to be friends, this essentially turns the graph into a social network and that is exactly the path the authors took to analyze the algorithm and methodology where groups are different ‘races’ of people in a community homeless shelter. Significance: The significance of this algorithmic methodology is very high as this paves the way for future researchers to adopt this approach since the proposed method is dataset independent. The implications of the embedding fairness constraints in a graph covering problem can be applied in epidemiology, fairness in media coverage, resource allocation, benefit disbursals in large scale corporations, only to name a few. This is possibly the strongest impact of the paper i.e. the openness of the approach to a variety of domains. Originality: Considering the originality, the paper is undoubtedly quite unique since it amalgamates a well-studied problem in computer science (graph covering) with group fairness constraints. This is further propounded by the authors since they very legitimately discuss the importance of this work using a social network dataset of a homeless shelter which has multiple races, genders etc. and show how the node coverage of such groups fail to be fair using existing graph covering algorithms. In terms of impact of the real world, the authors also mention suicide monitoring, landslide assessment – that are of high significance in reality. Besides the applications, the algorithmic approaches are also truly original. What stands out is the analysis of the problem where the authors first solve the problem of unfair node coverage in a social network by imposing a fairness constraint (by specifying that a fixed fraction of members of every group must be covered i.e. a lower bound – termed as RCfair in the paper) and then further analyze the problem to transform it into an MILP for the simplicity offered by Bender’s decomposition (termed as Theorem 1 in the paper). A laudable work, in its own regard is the solution approach of a robust optimization approach for fair graph coverage i.e. Problem 3 using a two-step method. Then the authors reformulate it into Problem 4, which is a K-adaptability counterpart problem. But the authors do not stop there and simply solve the max-min-max optimization subject to several constraints; rather they take another leap to analyze it again to reformulate the same problem further into a moderately sized MILP. Therefore, the authors take three formulations on the same problem – which effectively shows the depth of analysis from the researchers. This is praiseworthy. To be more specific, the final reformulation (Theorem 1) i.e. formulation of an MILP makes the solution of the problem much simpler (if not fully tractable) and using the observation of symmetry in the solution space, the authors present a practically feasible solution to the problem presented. Clarity: In terms of clarity, the authors have handled a complex problem in a very clear and comprehensible manner from the very beginning of the Abstract till the end. There are some minor caveats to the clarity which will be discussed in detail in Part 5 (Improvements) of the review. The authors introduced the problem in plain terms, before moving on to explicitly discuss the impact and application of the theoretical aspects of the research before delving in the algorithmic detail. This approach in clarity is exemplary. More paper should be like this. It should be noted that the clarity and simplicity of the language at the beginning of the paper do not take away any merit of the algorithmic and theoretical work presented in great detail in the latter part of the paper including the Appendix. The organizational flow of the paper is also as vivid as it could be, given the complexity of the problem that is discussed. Right after the introduction and literature review, the authors presented the formal formulation of the problem i.e. fair and robust graph covering problem. First, the authors formally introduced the problem where no fairness constraint is involved in the graph covering scenario as an optimization problem (called Problem RC in the paper). Then, the authors moved on to define the problem that is unique to this paper, which takes the group fairness into account (termed as RCfair). After formally formulating the problems and right before presenting the algorithmic solutions and approaches, the authors also defined the Price of Group Fairness (PoF) in Section 3 – which was aptly put in the right place. Here, authors discuss two type of PoF – deterministic case and uncertain case. In the first case, the authors assume that all monitoring (covering) nodes will survive whereas in the latter (uncertain) case, the authors defined the problem so that a particular number of nodes will fail with a given probability distribution. Both the cases are clearly defined in equation 2, Proposition 1 and Proposition 2. After all the formulations have been completed at this point, the authors moved on to the core solution approach(es) in section 4. Here, it should be noted that the authors should have indicated that the formulation of PoF and an extensive analysis of Section 2 of the paper is also presented in the Appendix. This would let the reader know that the theoretical proofs are also available. A list of all rooms for improvements are enumerated in Section 5 of the review. It should also be noted that the authors very clearly discussed any and all assumptions that has been made. For example, in Assumption 1, the authors present the average complexity of the probability of the existence of an edge from one to another among the same group as the inverse of the size of the community. The author(s) also discuss the reasoning for their assumptions. For example, the reasoning for Assumption 1 (mentioned above) was due to the “homophily” phenomenon in sociology. This is splendid in terms of clarity and transparency from the author(s). However, there were minor instances where there are some improvements of clarity. One such example is the range of lines from 280 to 283. It is not clear as to why the time limit was set to 2 hours. Was it for practical purposes or something else? It would have been better if it was clarified. The Appendix section is praiseworthy because it tackles every proposition, lemma and proof presented in the original paper. It is also done in great attention to detail. While there are some minor rooms for improvement, the Appendix is splendid. Some more rooms for improvement on clarity is given in section 5 (Improvements) of the review. Quality: The algorithmic formulations are very clearly presented – from problem formulation to reformulation into different schemes and the final solution. At the very start of the solution approach, the authors very clearly state that the solution will be done in three stages and also provides rational behind it i.e. the discontinuity of the objective function and the fact that it is non-linear, which they later reformulated into a linear problem by relaxing some constraints without losing the effective merit and significance. The paper has impressive intellectual merit in the reformulation of the original problem using sound mathematical proofs and equations. For instance, in section 4, the authors reformulated the original problem into a two-stage robust problem in problem 3. Author(s) also showed a brief proof showing the equivalence of the original problem (RCfair) and problem 3 (which is a linear objective function). Next, the reformulated problem (problem 3’s objective function) is turned into a K-adaptability counteract problem and the authors generated K number of candidate covering schemes in the first stage and after revelation of the nodes that indeed does not fail (which is not known apriory), the best candidate is chosen. All of these processes are very clearly defined and expressed in detail in high quality of intellectual merit. Then the authors presented possibly the most notable contribution to the problem i.e. the reformulation of the original RCfair problem into a moderately sized MILP. This follows from the previous reformulation of the K-adaptability problem (problem 4) – hence the flow remains coherent in the paper. This is presented using Theorem 1. Even though the mathematical formulation is clear and correct, it would have served well in terms of clarity to define each and every coefficients presented in Theorem 1. As such, a short explanation of the variables and coefficients in Theorem 1 in 2-3 sentences would remove the necessity to move to go on to the Appendix and the codes since the paper is so well written otherwise. Following the reformulation into an MILP, the authors very clearly the utility of Bender’s decomposition in solving the problem. Moreover, one of the best contributions and observations of the paper is to notice that the solution space has a very useful symmetry i.e. all permutations of a set of feasible covering schemes are also feasible. They used this information to tackle the combinatorial explosion of solutions. Finally in the results section, firstly, the performance (coverage) of the approach is compared with several other approaches which showed its superiority in the cases where the candidate schemes (K) were 2 and 3 vs. Greedy approach and DC approach (along the lines of 17% and 24% respectively). Secondly, the authors compared the solver time between standard Bender’s decomposition and Boosted Bender’s decomposition (using the available information of symmetry to relax the problem). They showed significant speedup in convergence towards feasible solutions holding one group constant while varying the size of another group. In a different metric (Table 2), the authors showed the improvement of their approach in terms of coverage percentage along with PoF over the same dataset that was introduced in the motivation i.e. a social network of a homeless shelter as they varied the number of nodes that have an uncertainty level of failing, compared to exact optimal coverage and greedy heuristics. Hence, the results were clearly illustrated to portray the strength of the work. It should be noted that the author(s) very honestly concur that in certain cases, the memory ran out which is reasonable. Even though it is understandable that the first formulation is intractable, there could have been a brief explanation of this along with the reasoning for memory overflow for K=2,3.

Reviewer 3



This paper formulates the problem of robust graph covering with fairness constraints. Put simply, this problem is to choose a set of nodes to maximize the number of nodes covered by this set (where 'covered' means adjacent to a node in the set) under the assumption that some number of nodes in the cover will fail (hence the 'robustness'), subject to the constraint that the fraction of each group (e.g. race, gender, other protected status) covered is similar. The authors define the price of fairness (PoF) and show that it can be unbounded in general, but they can bound a relaxation of PoF (i.e. the ratio of expected nodes covered in the fair vs expected nodes covered in the optimal policy rather than expectation of the ratio) for the stochastic block model. Actually computing the optimal fair allocation is NP hard, so the authors instead propose an approximation algorithm. The idea is to show that the original problem is equivalent to a k-adaptability problem under certain conditions, which can then be formulated as a mixed integer linear program and solved via Benders' decomposition (using symmetry to mitigate combinatorial blowup in the number of policies). I believe this is the first formulation of this problem, though it bears similarity to the fair influence maximization problem (which the authors cite as [31]). Since there are real-world instances of graph cover algorithms being used to deploy social intervention, incorporating fairness constraints is an important innovation and may impact people's lives, especially since the authors demonstrate that failing to previous SotA algorithms without fairness constraints can produce unfair solutions. For the most part, the writing is clear, but much of the interesting and important proofs are in the appendix, hiding how the results are achieved. The paper appears technically sound, but the in-body work hides much of the detail. In particular, Section 4 is not easy to follow. I am not sure how to remedy that, as the proofs and ideas are quite technical and detailed. Still, it would be nice to add proof sketches of e.g. Prop 1 and 2 and possibly Prop 3 and the reformulation in Theorem 1. The experimental evaluation is useful, but could use some additional description as suggested below. Questions/suggestions: - Your formulation requires that individuals have only one protected attribute, right? That seems to be implied by your assumption that the groups are disjoin but I think this should be made explicit. - I think it would be good to explicit contrast with [31], as fair influence maximization on networks seems like quite a related problem - It would be nice to have some discussion of the form of the worst case network in the body rather than leaving it to the appendix. - It would be nice to explain why you relax of PoF to ratio of Expected outcomes rather than expectation of ratios. (And if the answer is 'it's analytically tractable', that's better than not addressing it, in my opinion.) - In Prop 1 and 2, you mean that the worst-case \bar{PoF} Occurs when I is such and such, right? That's what the text suggests, but I don't see where you state that this is an upper bound over possible choices of I. - How should the reader interpret the upward closure property? I.e. when would it not be satisfied? - Is there any guidance on choosing k reasonably? - Is there no way to compress Theorem 1? I think it is overwhelming to the reader, and seeing the particular forms doesn't convey much insight. - Table 2 was quite hard to interpret. Several issues: first, the point that the 0 1 2 3 4 5 0 1 2 3 4 5 row refers to levels of J was not obvious, even though the title is 'Uncertainty Level J', because that text is too far from the row, and the row doesn't have label in line with the values. Second, the fact that the 0 in both metrics is with respect to the optimum, but the 1,2,3,4,5 is with respect to the greedy heuristic, is not at all indicated in the table, only the caption, and the caption is hard to parse (I suggest a slight rewrite: 'exact optimal coverage for J=0 and greedy heuristic for J>0' - just moving the labels closer to the text describing them). Third, it PoF figures are presented as percentages, at least if I understand the text correctly, but previously we have discussed PoF as ratios. initially I was confused, since the PoF as defined is between 0 and 1 but all these numbers are larger. Fourth, there are two bold numbers that don't seem to be explained - why are these numbers bold and why are they the only bold? Finally, the word 'Size' didn't immediately register as number of nodes for me. I would just call it Nodes or N. - Re Benders' decomposition, as I am not particularly familiar with it, are there any guarantees? The authors state that the 'iterative process continues until a cut-off time is reached'. In practice, does this mean the algorithm can fail, and if so, can the authors speak to when/how often/under what conditions this happens? - In the experiments, it would be nice to have more discussion about your resource assumptions. I.e. if N is 182, and you have I=N/3, that's 60 nodes, so J=5 is assuming that less than 10% of the monitors can fail. Is there any reason to think this is reasonable in that settings? Is I=N/3 reasonable? Also, you never explicitly wrote gamma, and N is not stated in the text, so it is hard to judge how large these monitor resources are meant to be. Typos: - pg 5 proposition 2 'If I=' - pg 6 'which decide' - pg 7 'adaptabaility' Update after feedback: I thank the authors for their responses to my questions, and am glad to hear that the authors have incorporated addressed many of my recommendations. I am not sure whether the authors meant to indicate that points II and X would or would not also be incorporated into the paper, but I do suggest they incorporate them (even if with the brevity in this rebuttal).

[Author Response · NeurIPS 2019]

We sincerely thank all of you for the detailed, thoughtful, and constructive comments and feedback. We have incorporated all your suggestions in our paper, which has significantly improved from them. We elaborate below.

**Reviewer #2: (I)** Our algorithm can handle $> 2$ *protected groups*: in our numerical results, there are up to five protected (racial) groups. It can also handle $> 2$ *protected attributes* (e.g., race, age, gender) by either: a) partitioning the network based on joint values of the protected attributes, and imposing max-min fairness constraint for each group; e.g., constraints on (White, Young, Female) people, etc.; or b) imposing a max-min fairness constraint for each protected attribute, separately. **(II)** We added a table of racial composition data for all networks. For instance, the MFP networks consisted of 16.5% White, 35% Black, 21% Hispanic, 18.5% Mixed, and 9% Others. Each individual belongs to a single race level. **(III)** The computational complexity of the problem increases exponentially with $K$, limiting us to increase $K$ beyond 3 for the considered instances. As demonstrated by our results, $K \sim 3$ was sufficient to considerably improve fairness of the covering at moderate cost. **(IV)-(V)** Covering schemes are not inputs but rather *decision variables* of the $K$-adaptability problem. The optimization problem will identify the best $K$ covering schemes that satisfy all the constraints including fairness constraints. **(VI)** In Section 5, we vary $W$ from 0 to 1, in increments of 0.04; we employ the largest $W$ for which the problem is feasible (see lines 152-154). By construction, this choice of $W$ guarantees that all of the fairness constraints are satisfied. The choice of $W$ varies with the network structure, no. of monitors, no. of failed nodes and $K$. In Table 2, for network MFP2, and for $J = 1, \ldots, 5$, $W$ was: 0.64, 0.56, 0.48, 0.4 and 0.32, respectively. We will report the values of $W$ in a table in the appendix. **(V)** We clarify Figs. 2(b)-(d) by changing the y-axis label to "Average normalized objective value" and adding to the caption that "it corresponds to the ratio of objective value of the master problem (appendix, line 582) to the network size, averaged over the five network instances."

**Reviewer #3 (I)-(II)** We incorporated all the recommendations. We included proof sketches, in the main text, after Props. 1 and 2 and Th. 1. We improve clarity of Th. 1 by adding "In this formulation, there are two sets of variables: a) The decision variables of the original problem; b) Dual variables emerging from employing linear programming duality to reformulate the inner minimization problem in Problem (4) ". We explain the role of the dual variables, and the two sets of constraints corresponding to different values of the parameter $\boldsymbol{l}$. **(III)** The memory overflow is due to the fact that the MILP formulation in Th. 1, although polynomial in all problem inputs, remains exponential in $K$. This is the main motivation to develop the Bender's decomposition approach in Section 4. Please see also response (III) to Reviewer #2. **(IV)** We will provide a head-to-head comparison with Table 1. For instance, the corresponding results of our approach ($K = 3$) for MFP2 network are: White: 56%; Black: 80%; Hispanic: 70%; Mixed: 71%; Other: 72%. **(V)** We improved the $K$-adaptability formularization by adding to Section 4: "the MILP reformulation relies on three key components: a) partitioning of the uncertainty set (achieved by introducing the parameter $\boldsymbol{l}$), b) continuous relaxation of each subset of the uncertainty set , and c) linear programming duality theory, to reformulate the robust optimization formulation over each subset." **(VI)** We will release the code and a "readme" file with instructions, detailing the sequence of the runs. **(VII)** The 2-hour time limit is justified by the "flattening" in the "Objective Bound" (Figures 2(b)-(d)); this is a common approach in optimization to terminate the algorithm when the change in objective is small. **(VIII)** We apologize for the confusion caused by Line 281. We now write "...by imposing fairness constraints for each group. We set the number of monitors to I=N/3." Please see also our answer (VI) to Reviewer #2. **(IX)** We now add a section on future work. **(X)** The Bernoulli distribution of the random variables $Y_n$ and $Z_{ni}$ is due to the Erdős-Rényi network generation process (see lines 418-419). Therefore, the probability of $Z_{ni}$ (similarly $Y_n$) taking the value of 1 is a known constant. **(XI)** The "budget regime" refers to the assumptions on the values of $I$, which we made more explicit. **(XII)-(XIII)** The remaining comments were addressed; we also added a part that was inadvertently deleted in the proof of Prop. 2.

**Reviewer #4 (I)** Please see answer (I) to Reviewer #2. **(II)** The paper [31] does not handle the uncertainty in node availability, which is one of the main contributions of our framework. **(III)** We have added the discussion of the worst-case PoF (Lemma 2) to the main text. **(IV)** We clarified, in the text, that we investigate the ratio of expected coverage rather than expectation of ratios for analytical tractability. **(V)** The assumption on $I$ can be interpreted as a "small budget assumption" that helps simplify the evaluation of the coverage. Please also see answer (II) to reviewer #3. **(VI)** Intuitively, uncertainty sets involving constraints as lower bounds on the (sums of) uncertain parameters satisfy the upward-closeness property. We now provide three examples of such sets, that are of practical relevance.**(VII)** The value of $K$ determines the approximation quality, enabling the decision-maker to trade-off the optimality with computation time. The choice of $K$ is mainly guided by the available computational resources (e.g., time) and is domain specific. Particularly, in low-resource settings (e.g., suicide prevention for homeless youth), we may be restricted to use low values of $K$. **(VIII)** Please refer to answer (II) to Reviewer #3. **(IX)** We have incorporated all your comments to improve the interpretability of Table 2. **(X)** Bender's decomposition is an *exact* iterative algorithm that converges to an optimal solution provided subproblems are LPs as in our case (Bertsimas, Dimitris, John N. Tsitsiklis. Intro. to linear optimization, 1994). In practice, it is run until a termination criterion, such as time, optimality gap, etc. is satisfied. We chose time limit for practical purposes. **(XI)** From discussion with our social work partners, $I \in [20, 30]\%N$ is typically seen in the context of suicide prevention. We now have added more rows in Table 2, reporting the average coverage improvement and average PoF for different values of $I$. For instance, for $I = 20\%N$, the average "Improvement in Min. Fraction Covered" for $J = 0, \ldots, 5$ is 17.2%, 13.8%, 14.0%, 10.0%, 9.0% and 6.7%, respectively. The "PoF" values are all less than 4%. The value of $\gamma$ can be inferred from $J$ ($\gamma = J/I$), we replace "Size" with $N$ in Table 2 for clarity.

[Meta-Review · NeurIPS 2019]

The reviewers agreed that the paper presents a novel intersection of ideas from disparate fields, identify a potentially critical fairness problem for graph-covering algorithms, then propose corrective methods with theoretical analysis. The author response helped clarify some misunderstandings, and all reviewers ended up in favor of the paper appearing at NeurIPS.